



# Application of a Satellite-Retrieved Sheltering Parameterization (v1.0) for Dust Event Simulation with WRF-Chem v4.1

Sandra L. LeGrand[1,2], Theodore W. Letcher[3], Gregory S. Okin[2], Nicholas P. Webb[4], Alex R. Gallagher[3], Saroj Dhital[4], Taylor S. Hodgdon[3], Nancy P. Ziegler[3], and Michelle L. Michaels[3]

[1]U.S. Army Engineer Research and Development Center, Geospatial Research Laboratory, Alexandria, Virginia, USA
[2]Department of Geography, University of California, Los Angeles, California, USA
[3]U.S. Army Engineer Research and Development Center, Cold Regions Research and Engineering Laboratory, Hanover, New Hampshire, USA
[4]USDA-ARS Jornada Experimental Range, Las Cruces, New Mexico, USA
**Correspondence:** Sandra LeGrand (Sandra.L.LeGrand@usace.army.mil)

**Abstract.** Roughness features (e.g., rocks, vegetation, furrows, etc.) that shelter or attenuate wind flow over the soil surface can considerably affect the magnitude and spatial distribution of sediment transport in active aeolian environments. Existing dust and sediment transport models often rely on vegetation attributes derived from static land-use datasets or remotely-sensed greenness indicators to incorporate sheltering effects on simulated particle mobilization. However, these overly simplistic ap-

proaches do not represent the three-dimensional nature or spatiotemporal changes of roughness element sheltering. They also ignore the sheltering contribution of non-vegetation roughness features and photosynthetically-inactive (i.e., brown) vegetation common to dryland environments. Here, we explore the use of a novel albedo-based sheltering parameterization in a dust transport modeling application of the Weather Research and Forecasting model with Chemistry (WRF-Chem). The albedo method estimates sheltering effects on surface wind friction speeds and dust entrainment from the shadows cast by subgrid-scale

roughness elements. For this study, we applied the albedo-derived drag partition to the Air Force Weather Agency (AFWA) dust emission module and conducted a sensitivity study on simulated $PM_{10}$ concentrations using the Georgia Institute of Technology-Goddard Global Ozone Chemistry Aerosol Radiation and Transport (GOCART) model as implemented in WRF-Chem v4.1. Our analysis focused on a convective dust event case study from 3-4 July 2014 for the southwest United States desert region discussed by other published works. Previous studies have found that WRF-Chem simulations grossly overesti-

mated the dust transport associated with this event. Our results show that removing the default erodibility map and adding the drag parameterization to the AFWA dust module markedly improved the overall magnitude and spatial pattern of simulated dust conditions for this event. Simulated $PM_{10}$ values near the leading edge of the storm substantially decreased in magnitude (e.g., maximum $PM_{10}$ values reduced from 17,151 $\mu$g m$^{-3}$ to 8,539 $\mu$g m$^{-3}$), bringing the simulated results into alignment with the observed $PM_{10}$ measurements. Furthermore, the addition of the drag partition restricted the erroneous widespread dust

emission of the original model configuration. We also show that similar model improvements can be achieved by replacing the wind friction speed parameter in the original dust emission module with globally-scaled surface wind speeds, suggesting that a well-tuned constant could be used as a substitute for the albedo-based product for short-duration simulations where surface roughness is not expected to change and for landscapes where roughness is constant over years to months. Though this alter-



native scaling method requires less processing, knowing how to best tune the model winds *a priori* could be a considerable

challenge. Overall, our results demonstrate how dust transport simulation and forecasting with the AFWA dust module can be improved in vegetated drylands by calculating the dust emission flux with surface wind friction speed from a drag partition treatment.

## 1 Introduction

Surface roughness features such as rocks, vegetation, and soil ridges created by tillage attenuate wind flow over the soil surface,

considerably affecting aeolian sediment transport and dust emission patterns (see reviews by Mayaud and Webb, 2017; Shao et al., 2015). Representing aerodynamic roughness effects and their spatiotemporal dynamics effectively in numerical dust models is, therefore, critical for estimating and forecasting the spatial patterns, timing, magnitude, and frequency of mineral dust emission accurately (Evans et al., 2016; Fu, 2019; Ito and Kok, 2017; Li et al., 2013; Tegen et al., 2002; Webb et al., 2014a, b). Several methods are available for characterizing vegetation effects in dust emission models (e.g., Evans et al.,

2016; Ginoux et al., 2001; Ito and Kok, 2017; Kim et al., 2013; King et al., 2005; Koven and Fung, 2008; Marticorena and Bergametti, 1995; Okin, 2008; Raupach et al., 1993; Tegen et al., 2002). However, many of these techniques require in situ data that are challenging to obtain over large spatial footprints or incorporate parameterizations with dependencies on land use or vegetation datasets that are often out of date or fail to represent the complex three-dimensional heterogeneity of landscape roughness elements (e.g., Pierre et al., 2014; Raupach et al., 1993).

Tuning dust models to satellite- and ground-based observations of dust concentrations and dust optical depth have generally been sufficient approaches for making dust models useful for large-scale climate and air quality applications (e.g., Cakmur et al., 2006). These particular applications tend to focus on total atmospheric dust aerosol loading trends instead of individual dust events. If the goal is to capture the general magnitude and frequency of large-scale dust aerosol patterns, tuning practices are often a viable solution to simulated dust emission errors. However, the relatively poor representation of surface roughness

effects on wind erosivity in current dust models has limited our capacity to forecast individual dust events accurately in desert regions with vegetation. Furthermore, poor roughness effect representation limits our ability to accurately simulate the influence of land management practices and desertification on spatial and temporal patterns of dust emission (Webb and Pierre, 2018).

Sediment mobilization schemes are often represented in terms of wind friction speed, $u_*$, a scalar parameter commonly used to describe processes related to wind shear stress. Near the land surface, $u_*$ represents the total wind shear stress acting on both

the roughness features and the immediate soil surface (Raupach, 1992; Raupach et al., 1993). As a result, we can divide $u_*$ into roughness ($u_{r*}$) and soil surface ($u_{s*}$) components (generally termed drag partitioning). The wind shear stress that reaches the immediate soil surface governs particle mobilization, so dust emission models driven by $u_{s*}$ (or wind erosivity) instead of $u_*$ may produce better outcomes (e.g., Darmenova et al., 2009; Okin, 2008; Webb et al., 2020). However, this assumption is worth testing for individual modeling applications since more physically sophisticated methods do not always lead to better

simulation outcomes, especially when the more advanced modeling approaches are limited by uncertainties in their required input parameters (e.g., LeGrand et al., 2019).



Though roughness elements affecting $u_{s*}$ could comprise any ground feature obstructing airflow over the land surface, traditional aeolian transport models generally equate roughness to vegetation cover, with varying degrees of sophistication in their approach. For example, some models restrict or reduce dust fluxes based on static prescribed vegetation or land use
attributes (e.g., Ginoux et al., 2001; LeGrand et al., 2019; Woodward, 2001). Others implement dynamic masks or spatially-varying dust flux scaling factors based on real-time or climatological datasets derived from greenness fraction, Leaf Area Index, or Normalized Difference Vegetation Index satellite data (Asadov and Kerimov, 2019; Collins et al., 2011; Evans et al., 2016; Ito and Kok, 2017; Kim et al., 2013; Kok et al., 2014; Solomos et al., 2019; Tegen et al., 2002; Vukovic et al., 2014). While some of these techniques have proven useful in certain modeling applications, they depend on datasets that primarily highlight
plant productivity phases (e.g., Yu et al., 2016), which may not translate well to soil surface sheltering (Okin, 2010).

Chappell and Webb (2016) proposed a method for parameterizing $u_{s*}$ in aeolian transport models using remotely-sensed surface albedo ($\omega$). Their technique infers the drag partition from shadows ($1-\omega$) cast by roughness elements based on the assumption that shadows can serve as a proxy for the sheltered surface area extent (Chappell et al., 2010). As described in Ziegler et al. (2020), a generalizable form of the Chappell and Webb (2016) approach equates to:

$$u_{s*} = u_{ns*} U_{10\mathrm{m}}, \tag{1}$$

where $U_{10\mathrm{m}}$ is the wind speed 10 m above ground level, and $u_{ns*}$ is a normalized $u_{s*}$ parameter (unitless) derived from albedo. By using an albedo-based calculation for each pixel area, the Chappell and Webb (2016) method provides an areal estimate of $u_{s*}$ for both the sheltered and non-sheltered zones of the soil surface within each grid pixel.

Here, we explore the use of the Chappell and Webb (2016) albedo-based drag partition within the Weather Research and
Forecasting model with Chemistry (WRF-Chem). WRF-Chem is a physically-based Earth-system model that simulates atmospheric motion on a non-hydrostatic, Eulerian grid in addition to the emission, transport, and mixing of gases and aerosols simultaneously with the meteorology (Fast et al., 2006; Grell et al., 2005; Peckham et al., 2017). The WRF-Chem framework is configurable and can be run with a variety of aerosol and atmospheric chemistry parameterizations, depending on a user's specific interests and computational resources. For this effort, we chose to use the AFWA dust emission module (LeGrand
et al., 2019) with the Georgia Institute of Technology–Goddard Global Ozone Chemistry Aerosol Radiation and Transport (GOCART) model (Chin et al., 2000; Ginoux et al., 2001) as implemented in WRF-Chem v4.1.

The AFWA dust emission equations assume wind-driven dust entrainment primarily occurs through a process called saltation bombardment, in which wind-lofted sand-sized particles ($\sim$50 – 2000 $\mu$m diameter) too heavy to remain suspended in the air collide with the land surface and eject smaller dust-sized particles (generally < 20 $\mu$m diameter) upon impact (e.g., Gillette,
1977; Kok et al., 2012). Several studies have investigated the use of the AFWA dust emission module for a variety of dust modeling applications (e.g., Aragnou et al., 2021; Francis et al., 2022; Hamzeh et al., 2021; Karumuri et al., 2022; Kuchera et al., 2021; Mesbahzadeh et al., 2020; Miller et al., 2021; Mohebbi et al., 2019; Péré et al., 2018; Saidou Chaibou et al., 2020; Solomos et al., 2018; Teixeira et al., 2016; Tsarpalis et al., 2018, 2020; Uzan et al., 2016; Xu et al., 2017; Zhang et al., 2022; Zhou et al., 2019). In general, these studies highlight useful applications of the model. However, several authors noted the





need for improved dust source and land surface characterizations in regions with heterogeneous terrain (Cremades et al., 2017; Fountoukis et al., 2016; Hyde et al., 2018; Kim et al., 2021; Ma et al., 2019; Mohebbi et al., 2020; Nabavi et al., 2017; Nguyen et al., 2019; Nikfal et al., 2018; Parajuli et al., 2019, 2020; Parajuli and Zender, 2018; Rizza et al., 2016, 2021; Spyrou et al., 2022; Su and Fung, 2015; Yuan et al., 2019; Zhao et al., 2020).

Here, we aim to evaluate the sensitivity of dust transport simulated by WRF-Chem to the Chappell and Webb (2016) albedo-based drag partition. Our analysis focused on a convective dust event case study from 3-4 July 2014 for the desert southwest U.S. region previously discussed by other published works (e.g., Hyde et al., 2018; Yu and Yang, 2016). The results from Hyde et al. (2018), in particular, motivated our case study choice. They found that the AFWA dust emission scheme used in WRF-Chem grossly overestimated dust transport for this event. We hypothesized that better geographic sheltering-effect representation from the Chappell and Webb (2016) drag partition would improve the overall magnitude and spatial pattern of the emitted dust and ultimately result in an improved simulation of transported dust.

## 2 Methodology

### 2.1 Dust emission flux calculation

The AFWA dust emission module (LeGrand et al., 2019) is an adaptation of the dust emission scheme originally described by Marticorena and Bergametti (1995). Equations comprising the AFWA code are in terms of $u_*$ and primarily include the threshold friction speed required for particle entrainment over an idealized smooth surface ($u_{*ts}$), the horizontal saltation flux ($Q$), the bulk vertical dust flux ($F_B$), and the size-resolved emitted dust flux. For a detailed overview of the dust emission module equations, see LeGrand et al. (2019). The following two sub-sections summarize the main components of the AFWA dust emission model and the modifications required to incorporate the Chappell and Webb (2016) drag partition.

#### 2.1.1 The original AFWA module configuration

In the AFWA dust emission calculation, soil particles are divided into a predetermined number of bins based on their effective particle size (referred to as size bins). Tables 1-2 provide attributes associated with the nine size bins used for saltation-based processes and the five size bins used for emitted dust. Here, we denote effective diameters for saltation and dust particles by $D_{s,p}$ and $D_{d,p}$, respectively.

As the simulation evolves, saltation for a given size bin initiate and cease as $u_*$ exceeds or falls below size-resolved values of $u_{*ts}$, respectively. First, the module generates a semi-empirical $u_{*ts}$ estimate (in units of centimeters per second) for each saltation size bin, $u_{*ts}(D_{s,p})$, assuming air-dry soil conditions:

$$u_{*ts}(D_{s,p}) = 0.13 \frac{\left(\frac{\rho_{s,p} g D_{s,p}}{\rho_a}\right)^{0.5} \left(1 + \frac{0.006}{\rho_{s,p} g D_{s,p}^{2.5}}\right)^{0.5}}{\left[1.928 \left(a_{mb}(D_{s,p})^x + b_{mb}\right)^{0.092} - 1\right]^{0.5}},$$

(2)





where g = 981 cm s$^{-2}$ is the gravitational constant, $\rho_a$ is the spatiotemporally varying air density from the lowest model atmospheric layer, $\rho_{s,p}$ is the particle density of saltation bin $p$, $x = 1.56$, $a_{mb} = 1331$ cm$^{-x}$, and $b_{mb} = 0.38$. The code then applies

a correction function, $f(\theta)$, that incorporates soil water content and clay composition to account for the effects of soil moisture on particle cohesion based on the approach of Fécan et al. (1999):

$$u_{*ts}\left(D_{s,p},\theta\right) = u_{*ts}\left(D_{s,p}\right)f\left(\theta\right). \tag{3}$$

Next, the module diagnoses size-resolved saltation flux ($Q(D_{s,p})$: in units of grams per centimeter per second) values for each saltation size bin following,

$$Q\left(D_{s,p}\right) = \begin{cases} C\frac{\rho_a}{g}u_*^3\left(1 + \frac{u_{*ts}(D_{s,p},\theta)}{u_*}\right)\left(1 - \frac{u_{*ts}(D_{s,p},\theta)^2}{u_*^2}\right), & u_* > u_{*ts}\left(D_{s,p},\theta\right) \\ 0, & u_* \le u_{*ts}\left(D_{s,p},\theta\right) \end{cases}, \tag{4}$$

where $C$ is an empirical proportionality constant set to 1.0. The module then multiplies $Q(D_{s,p})$ by bin-specific weighting factors ($dS_{\mathrm{rel}}$) determined from prescribed mass distribution assumptions and spatially varying sand, silt, and clay mass fractions. The integrated sum of the weighted $Q(D_{s,p})$ values determines the total streamwise saltation flux, $Q$, associated with each model grid cell:

$$Q = \sum_{s,p=1}^{9} Q\left(D_{s,p}\right) dS_{\mathrm{rel}}\left(D_{s,p}\right). \tag{5}$$

After determining $Q$, the module generates the bulk vertical dust emission flux ($F_B$; in units of grams per centimeter squared per second) triggered by saltation by multiplying the $Q$ field by a topographic-based dust source strength parameter ($S$) and a sandblasting efficiency factor ($\beta$; in units of value per centimeter) derived from the soil clay fraction. The $F_B$ calculation also includes an aerodynamic roughness length ($z_0$) conditional to limit dust emission to non-urban regions with relatively sparse

vegetation coverage:

$$F_B = \begin{cases} QS\beta, & z_0 \le 20\ \mathrm{cm} \\ 0, & z_0 > 20\ \mathrm{cm} \end{cases}, \tag{6}$$

where

$$\beta = \begin{cases} 10^{0.136(m_{\mathrm{clay}})-6}, & m_{\mathrm{clay}} < 0.2 \\ 1.06 \times 10^{-6}, & m_{\mathrm{clay}} \ge 0.2 \end{cases}, \tag{7}$$





and $m_{clay}$ is the soil clay mass fraction.

Effectively, the $m_{clay}$ conditional in Eq. (7) limits the sandblasting efficiency parameter to $1.06 \times 10^{-6}$ cm$^{-1}$ when the soil composition exceeds 20% clay content. However, as noted by LeGrand et al. (2019), the $\beta$ function primarily serves as a dimensional scaling factor. The overall effects of soil clay content variability on $F_B$ are relatively small.

The $S$ parameter (originally described by Ginoux et al., 2001) represents the availability of loose erodible soil material at a given location based on the degree of topographic relief of the surrounding area. This approach assumes that soil composition

remains consistent over time, and the simulated land surface will neither run out of dust material nor acquire new dust material through fluvial or atmospheric deposition as the simulation evolves. Essentially, $S$ is a spatially varying tuning parameter ranging from 0 to 1 that assumes erodible material accumulates in low points in the terrain, determined by:

$$S = \left( \frac{z_{\max} - z_i}{z_{\max} - z_{\min}} \right)^5, \tag{8}$$

where $z_i$ is the elevation of the cell, and $z_{\max}$ and $z_{\min}$ are the maximum and minimum terrain elevation in the surrounding

$10° \times 10°$ area, respectively. The AFWA module, however, does not directly diagnose $S$ using domain-relative elevation data. Instead, WRF-Chem interpolates static $S$ values initially derived from a $\frac{1}{4}°$ elevation dataset to the grid domain during the model preprocessing phase. Furthermore, this pre-built $S$ field incorporates a vegetation mask that blocks dust emission (i.e., $S = 0$) from areas designated as vegetation-covered according to a $1° \times 1°$ resolution 1987 annual average land cover dataset. While these settings may be appropriate for some modeling applications, the coarse nature of these input datasets likely limits

the spatial viability of S at mesoscale and convective-permitting model resolutions.

Next, the module applies a prescribed particle size distribution derived using the Kok (2011) brittle fragmentation theory to obtain size-resolved dust emission fluxes:

$$F\left(D_{d,p}\right) = F_B \kappa \left(D_{d,p}\right), \tag{9}$$

where $F(D_{d,p})$ is the size-resolved emitted dust flux (in units of grams per centimeter squared per second) and $\kappa_{d,p}$ is the

distribution fraction of the suspended dust size bin (see Table 2). Lastly, the AFWA module uses $F(D_{d,p})$ values to determine the size-resolved dust concentrations injected into the first model atmospheric level for transport during each model time step. From this point forward, functions from other modules in the GOCART suite take over processing the fate and transport of the airborne dust aerosols.

### 2.1.2   Incorporation of drag partitioning

To obtain gridded values of $u_{ns*}$ for use in WRF-Chem, we estimated surface shadowing using data from the snow-masked 500 m Moderate Resolution Imaging Spectroradiometer (MODIS) Bidirectional Reflectance Distribution Function (BRDF)





albedo daily product (Collection 6, MCD43A1). Following Chappell and Webb (2016) and Ziegler et al. (2020), we derived the normalized proportion of shadow (represented using a normalized albedo, $\omega_n$) by:

$$\omega_n = \frac{1 - \omega_{\mathrm{dir}}(0°)}{f\mathrm{iso}},$$ (10)

where $\omega_{\mathrm{dir}}(0°)$ is the daily nadir "black-sky" albedo for MODIS band 1 (620-670 nm wavelength) and $f_{\mathrm{iso}}$ is the band 1 BRDF isotropic weighting parameter. We then determined daily $u_{ns*}$ by:

$$u_{ns*} = 0.0311 \left( \exp \frac{-\omega_{ns}^{1.131}}{0.016} + 0.007 \right),$$ (11)

where $\omega_{ns}$ represents an empirically-scaled proportion of shadow obtained via:

$$\omega_{ns} = \frac{(a - b)(\omega_n - 35)}{-35} + b,$$ (12)

using the scaling factors $a = 0.0001$ and $b = 0.1$ to align $\omega_{ns}$ with the ray-casting performed on the reconstructed roughness element configurations in the wind tunnel study by Marshall (1971).

To incorporate $u_{s*}$ into the AFWA dust emission module, we configured WRF-Chem to ingest daily MODIS-derived $u_{ns*}$ fields (Eq. (11)) that had been interpolated to the model grid domain and modified the dust emission equations to use $u_{s*}$ in place of $u_*$. Specifically, we converted daily $u_{ns*}$ values to instantaneous $u_{s*}$ estimates following Eq. (1), using the WRF-Chem

simulated wind speed at 10 m above ground level that updates each model time step as $U_{10\mathrm{m}}$. Finally, we replaced $u_*$ in the saltation function from Eq. (4) with $u_{s*}$:

$$Q(D_{s,p}) = \begin{cases} C\frac{\rho_a}{g}u_{s*}^3 \left(1 + \frac{u_{*ts}(D_{s,p},\theta)}{u_{s*}}\right)\left(1 - \frac{u_{*ts}(D_{s,p},\theta)^2}{u_{s*}^2}\right), & u_{s*} > u_{*ts}(D_{s,p},\theta) \\ 0, & u_{s*} \leq u_{*ts}(D_{s,p},\theta) \end{cases}.$$ (13)

Note that estimated dust emissions are particularly sensitive to small changes in wind friction speed due to the cubed component of the saltation equation (e.g., Tegen et al., 2002). This replacement of $u_*$ with $u_{s*}$ in Eq. (13) makes the saltation

equation consistent with the physics of aeolian transport in the presence of roughness. Essentially, this modification enables excess wind friction speed at the soil surface to govern the saltation mass flux rather than allowing the total wind friction speed acting over the entire roughness layer to drive saltation (Webb et al., 2020). In addition, we added model run-time configuration options to disable the $z_0$ conditional and remove the $S$ factor from the $F_B$ function (Eq. (6)) since both of these features incorporate broadscale vegetation masks. Readers are encouraged to review the report by Michaels et al. (2022) for a

detailed overview of the code implementation process and model runtime instructions.

The use of the snow-masked version of the MODIS product is essential for this particular modeling application. Although the AFWA module restricts dust emissions from snow-covered areas, the snow-covered grid spaces in the model may not align



with snow-covered pixels in the MODIS product. Accordingly, our model implementation process assumes $u_{s*} = 0$ m s$^{-1}$ for grid spaces with missing data in the MODIS product. This assumption effectively blocks dust emissions from MODIS-detected

snow-covered regions and water bodies.

### 2.2    Description of case study event

As described in detail by Gallagher et al. (2022), our case study dust event was driven by a convective outflow boundary associated with a thunderstorm cluster that developed and evolved over central and southern Arizona following convective weather patterns characteristic of the North American Monsoon's summer phase (e.g., Adams and Comrie, 1997). The event

began at 1800 UTC on 3 July 2014, peaked at around 0000 UTC on 4 July 2014, and concluded mainly by 1200 UTC on 4 July 2014. At the peak of the event, convective cells organized into an extensive north-south line, pivoting clockwise about its northern end. By 4 July 2014, 0200 UTC, the convective line collapsed into three components, with each portion progressing in different directions as they diverged. The gust front associated with the central cell just south of Phoenix, Arizona generated a thick wall of dust that preceded the storm as it continued west-northwest over the metropolitan area.

Our simulation analysis primarily focused on dust produced by the cell that affected Phoenix. However, we also chose this particular case study because dust concentrations outside the immediate Phoenix area were relatively low in magnitude or negligible. The model's ability to simulate minimal dust loading and dust-free (i.e., clear sky) conditions accurately is also important for it to be a reliable prediction tool. For the simulated results to be considered successful for this case study, the model should be able to both capture the progression of the dust wall over Phoenix and limit dust production from the rest of

the broader model domain.

### 2.3    Model configuration

WRF-Chem is a version of the Weather Research and Forecasting (WRF) model by Skamarock et al. (2019) with additional modules for atmospheric chemistry processes and feedbacks (Fast et al., 2006; Grell et al., 2005). Like WRF, WRF-Chem is a fully compressible finite difference model that simulates atmospheric motion on the Arakawa C-grid and incorporates a variety

of parameterizations for simulating sub-grid atmospheric motion, cloud microphysics, radiation, and terrain processes. We used WRF-Chem v4.1 for our test case simulation with WRF parent model configuration settings suggested by Gallagher et al. (2022) and chemistry settings from LeGrand et al. (2019) and Letcher and LeGrand (2018). Table 3 provides a brief overview of the model chemistry and physics configuration. However, complete pre-processor and run-time configuration files (referred to as the namelist.wps and namelist.input files, respectively) for this effort are available in the report by Michaels et al. (2022).

The model vertical grid used the default spacing distribution with 40 levels following a stretched hybrid sigma-pressure vertical coordinate that favors higher resolution near the ground. Figure 1 displays the three telescoping model domains (D01, D02, and D03, hereafter) with grid resolutions of 18 km, 6 km, and 2 km, respectively. In Fig. 2-3, we show key terrain attributes associated with the AFWA dust emission functions for D02 and D03, which primarily encompass the southwest U.S. desert region.



We set the WRF-Chem initial and lateral boundary forcing conditions using 3-hourly analysis fields from the Rapid Refresh
     Model (Benjamin et al., 2016) supplemented with 6-hourly soil moisture information from the National Centers for Environ-
     mental Prediction North American Model analysis product. Though these data are obtainable from multiple resources, we ac-
     quired all of our forcing datasets from the National Center for Environmental Information data portal: https://www.ncdc.noaa.gov/data-
     access/model-data/model-datasets (last access: 20 November 2019).

We performed our simulations for a two-day period from 3 July 2014, 0000 UTC to 5 July 2014, 0000 UTC (2 July 2014,
     1700 LT to 4 July 2014, 1700 LT for Arizona). The first 12 hours of the simulation (0000-1200 UTC 3 July 2014) were
     disregarded as spin-up to allow the model to adjust to the initial and lateral boundary conditions. Atmospheric dust was
     initialized using a "cold start" approach, which assumes an initial atmospheric dust concentration of zero. Settings engaged
     by the GOCART simple option in WRF-Chem generated atmospheric fields for other non-dust aerosols, including sea salt,
black carbon, organic carbon, and dimethyl sulfide. This particular configuration assumes background sea salt emissions based
     on the lowest model level wind speeds over the oceans (Gong, 2003) and emissions for the other three aerosol species using
     climatological emission datasets and the WRF-Chem PREP-CHEM-SRC preprocessing software (Freitas et al., 2011). All dust
     aerosols originated from the local model domain, meaning that no additional dust aerosols entered the D01 lateral boundaries
     as the simulations evolved. We consider this a reasonable assumption given the relatively short duration of the case study and
because the dust event generated from localized, convection-induced conditions well within the D01 boundaries.

     To better understand the sensitivity of the model to the drag partition treatment, we assessed five test configurations, including
     a simulation produced using the original AFWA code configuration without a drag partition as a control (CTRL) and four
     alternative versions using the modified size-resolved saltation treatment from Eq. (13) (see Table 4). The first alternative
     configuration (ALT1) applied the $u_{s*}$-based $Q(D_{s,p})$ estimates and the original $F_B$ calculation (Eq. (6)), which restricts dust
emission from grid cells classified by the model as areas with $z_0$ greater than 20 cm. Because we used the model default 21-class
     MODIS International Geosphere-Biosphere Programme (IGBP) dataset to characterize land use, this conditional effectively
     limits dust production to areas classified by the model as barren, grassland, shrubland, savanna, or cropland. The second
     alternative configuration (ALT2) is identical to ALT1 but with the $z_0$ conditional removed. Our third alternative configuration
     (ALT3) further simplifies ALT2 by removing the preferential source strength term from the $F_B$ calculation (i.e., $S = 1$). Lastly,
the fourth alternative configuration (ALT4) estimates $u_{s*}$ by applying a global scaling factor ($C_s$) to $U_{10m}$:

$$u_{s*} = C_s U_{10m}, \tag{14}$$

with $C_s$ set to a value within the general range of $u_{ns*}$ values estimated for the model domain by Eq. (11). Similar to ALT3, the
ALT4 configuration also ignores the the $z_0$ conditional and assumes $S = 1$.

     The ALT3 configuration tested the ability of the albedo-based approach to accurately represent the spatially varying surface
aerodynamic sheltering afforded by non-erodible roughness elements on the land surface. However, we specifically included
     the ALT4 configuration to test if the model is able to achieve similar results for this case study event by using globally
     scaled surface winds without the additional input data acquisition and preprocessing requirements inherent to the Chappell



and Webb (2016) method. In other words, ALT4 explores if a model user could achieve the same outcome as the Chappell and Webb (2016) drag partition scheme by simply tuning down the winds if the dust source areas in their domain of interest

were generally prone to coverage by vegetation or other roughness elements. Effectively, ALT3 and ALT4 are the only model configurations without some form of vegetation mask built into the dust emission code (see Fig. 4).

Figure 5 provides a comprehensive schematic summary of the five test configurations and their required input parameters. This multi-tiered testing approach enabled us to explore the drag partition's influence on simulation outcomes systematically. The $z_0$ dependency removal is relatively straightforward since it primarily serves as a vegetation mask. However, omitting

the $S$ parameter also removes the spatially varying available sediment supply tuning parameter from Eq. (6). Since the drag partition provides no direct information about sediment supply, it is not clear if removing $S$ is a universally valid approach. We note, however, that previous studies have found that the default WRF-Chem $S$ parameter provides a poor representation of dust source strength in our case study area (e.g., Vukovic et al., 2014; Parajuli and Zender, 2018). Setting $S$ to 1 forces the model to assume that all areas are functionally equally erodible and that dust emissions are solely a result of saltation and sandblasting

efficiency. Replacing the built-in WRF-Chem $S$ field without the 1987 AVHRR vegetation mask, though, is beyond the scope of this effort.

### 2.4  Observation data

We evaluated model performance using in situ observations from Automated Surface Observing Stations (ASOS). ASOS observations are available in standard Meteorological Aerodrome Report (METAR) format (e.g., WMO, 2018) every hour and

in the less conventional one-minute or five-minute format, depending on the station. Gallagher et al. (2022) previously assessed model wind speed for our simulation configuration in the vicinity of the main event convection using one-minute frequency ASOS data. Thus, our analysis primarily focused on comparing simulated data against ASOS present weather and visibility observations and wind speed measurements for ambient conditions away from the main convective dust event using ASOS data accessed through the Meteorological Assimilation Data Ingest System (MADIS) portal (https://madis.ncep.noaa.gov, last

access: 12 November 2021). We also compared simulated results against observed concentrations of particulate matter up to 10 $\mu$m in diameter (PM$_{10}$) using hourly PM$_{10}$ in situ measurements obtained from the U.S. Environmental Protection Agency's Air Quality System database (retrieved via an application programming interface at https://www.epa.gov/outdoor-air-quality-data, last access: 1 July 2020). We did not consult additional satellite-retrieved products like false color dust-enhanced imagery and aerosol optical depth to supplement our assessment due to cloud obscuration over the area of interest.

To evaluate simulated storm structure, location, and timing, we analyzed observed Next Generation Weather Radar (NEXRAD) composite imagery (available at https://www.ncdc.noaa.gov/nexradinv/, last access: 1 July 2020) to qualitatively compare and verify the location, structure, and intensity of the simulated convective storms. The radar composite images comprise a blend of radar reflectivity from individual locations with overlapping coverage to generate a cohesive radar map across the continental United States. This mosaic approach has the advantage of filling in blockages from individual radar sites with neighboring

ones, providing a more complete snapshot of convective activity across the radar network coverage area.



## 3 Results

### 3.1 Albedo-derived normalized surface friction speed

Figure 6 shows the normalized surface wind friction speed for the case study period diagnosed from MODIS BRDF data and Eq. (11) interpolated to the model domain. Although we consider $u_{ns*}$ to be a dynamic parameter, $u_{ns*}$ here is a static field (i.e.,

$u_{ns*}$ only varies spatially, not temporally) due to the relatively short duration of our case study simulation (48 hours). Note, the $u_{ns*}$ values in the areas characterized by the model as shrubland range from 0.0225 to 0.0325, barren areas are around 0.03 to 0.035, croplands are 0.0125 to 0.02, and forested and urban areas generally range from 0.005 to 0.0225. The upper limits of $u_{ns*}$ associated with forested and urban areas are likely higher than those of croplands due to roughness element height consistency and density. For example, plants in individual crop fields generally grow at relatively uniform rates and spacing,

giving the crop canopy a "smoother" aerodynamic roughness than urban centers and forested regions with more heterogeneity in roughness element height, shape, and density.

### 3.2 Environmental forcing conditions

We conducted a rigorous evaluation of the simulated environmental forcing conditions so we could account for any wind flow errors in our assessment of simulated $PM_{10}$ against hourly station-based $PM_{10}$ measurements. The model was able to reproduce

the storm's general structure and timing, including the formation of the initial quasi-linear convective system and the collapse of the convective line into individual cells. Furthermore, the simulated near-surface wind speeds were in good agreement with wind speeds observed at ASOS stations. However, simulated wind speeds peaked 1 to 2 hours early in some locations with slightly higher (about +1 m s$^{-1}$) intensity. According to Gallagher et al. (2022), these minor wind speed errors may be partly due to erroneous land use characterization, particularly in the higher terrain elevation areas where the storm initiated.

To better assess the potential influence of these minor errors in wind flow on simulated dust concentrations, we conducted a more in-depth review of the location and timing of the primary cell that produced the main dust event. Specifically, we examined locational differences between a central point along the storm's leading edge in both the observed- and modeled- radar reflectivity over the dust-producing cell's lifetime (Fig. 7). Overall, the simulated convective activity developed and propagated more northwest than the patterns we observed in the composite radar images. This divergence began with a slight

difference in where the cells initiated, followed by a more prevalent westward motion in the simulated conditions than the storm's actual southwest track. As the storm's evolution continued, the simulated event maintained its path and velocity while the observed event pivoted and accelerated. Though variation between the simulated and actual storm tracks occurred, this distance became smaller over time (Fig. 7c).

Due to these discrepancies, we must limit our simulated dust assessment to a qualitative evaluation of the overall storm

system rather than comparing hourly simulated time series of $PM_{10}$ to observed values associated with an exact geospatial point. This approach, however, is not inconsistent with convective weather analyses since simulating the precise placement and timing of individual convective cells is generally considered more luck than skill due to irreducible limitations in non-linear model physics, numerical processing, and initialization accuracy. Capturing the general spatial pattern and timing of





the dust event is the higher priority as model users can often correct persistent magnitude errors through global model tuning
parameters.

Simulated $U_{10m}$, $u_*$, and albedo-derived $u_{s*}$ values associated with the peak of the dust event (4 July 2014, 0000–0300 UTC) are provided in Fig. 8. Overall, the $U_{10m}$ patterns align well with the ASOS station observations and areas of convective activity seen in the radar composite imagery (see supplemental Fig. S1). We note that the simulated $u_*$ values are generally an order of magnitude stronger than their associated $u_{s*}$ partition. This outcome is likely due to the prevalence of shrubs and grasses in
southern Arizona and the more inland areas of southern California.

Figure 8 also shows the threshold friction speed for air-dry soil conditions, the soil moisture-based correction function, and the soil wetness-corrected threshold friction speed for a particle with a diameter of 69 $\mu$m ($u_{*ts}(D_{s,p} = 69\ \mu$m), $f(\theta)$, and $D_{s,p} = 69\ \mu$m, $\theta$), respectively). This particular particle size is the effective diameter for saltation size bin 7 (see Table 1). As described in a review paper by Kok et al. (2012, and references within), the $u_{*ts}$ parameter represents the minimum surface wind
shear stress needed to overcome the inertial, cohesive, and adhesive forces holding the particle to the soil bed. Interparticle forces (e.g., capillary, Van der Waals, electrostatic, chemical binding, and coulomb forces) keeping the particles fixed to the ground have a stronger effect on smaller particles due to their high ratio of surface area to volume. However, larger particles require more energy from the wind to mobilize because inertial force effects become stronger with increasing particle mass. The $u_{*ts}(D_{s,p})$ curve produced by Eq. (2) has a minimum of about 74 $\mu$m (assuming standard atmospheric pressure of 1225
kg m$^3$) because fine sand grains are less subject to interparticle forces than smaller clay- and silt-sized particles but are still relatively lightweight. Hence, fine sand-sized particles tend to be the first particles to mobilize from direct wind shear stress. Accordingly, particles associated with saltation size bin 7 ($D_{s,p} = 69\ \mu$m) in the AFWA module will be the first to mobilize. Thus, we consider spatial patterns related to saltation size bin 7 to be good diagnostic indicators of overall dust emission model behavior.

Air density is the only spatiotemporally varying parameter in the $u_{*ts}(D_{s,p})$ calculation. Though we can discern a reduction in $u_{*ts}(D_{s,p} = 69\ \mu$m) over time immediately under the convective line (Fig. 8m-p), the overall effect of air density on $u_{*ts}(D_{s,p})$ for this case is relatively negligible compared to the influence of soil moisture. These results also align with findings by Darmenova et al. (2009) in their assessment of the sensitivity of the Marticorena and Bergametti (1995) dust emission scheme to uncertainties in its required input parameters. Under air-dry soil conditions, $u_{*ts}(D_{s,p} = 69\ \mu$m) ranges between 0.17 to 0.19
m s$^{-1}$ across most of the domain. We also see relatively little change in the $f(\theta)$ field during the dust event, except for the area associated with a line of precipitation that occurred within the convective cell behind the main wall of dust (Fig. 8q-t). The $u_{*ts}(D_{s,p} = 69\ \mu$m, $\theta$) maxima over the Mogollon Rim aligns with isolated areas of convective precipitation that occurred earlier in the simulation (Fig. 8u-x). Note, however, that the $u_{*ts}(D_{s,p} = 69\ \mu$m, $\theta$) values along the Mogollon Rim adjacent to this maxima are around 0.2 to 0.3 m s$^{-1}$, comparable to the southwest Arizona region where the dust event occurred, and, for the
most part, are well below the simulated values of $u_*$.

Figure 9 shows the excess friction speed, $u_{*ex}$, for saltation size bin 7 associated with different treatments for particle mobilization during the peak of the event, where $u_{*ex}$ is the model simulated friction speed driving the saltation and dust emission equations (i.e., $u_*$ or $u_{s*}$) minus $u_{*ts}(D_{s,p} = 69\ \mu$m, $\theta$). The six examples provided in Fig. 9 include $u_{*ex}$ calculated





using the model-generated $u_*$ values, $u_{s*}$ values derived using the Chappell and Webb (2016) method (Eq. (1)), and four

instances of $u_{s*}$ values estimated from 10 m wind speeds scaled by a global tuning constant ($C_s$; Eq. (14)) as the wind forcing

parameter. We considered values of $C_s$ = 0.02, 0.025, 0.03, and 0.035 based on the general range of $u_{ns*}$ values diagnosed for

the area where the dust event occurred (e.g., Fig. 6; note, $u_{ns*}$ has a theoretical maximum of 0.0381). Effectively, positive $u_{*ex}$

values indicate that wind friction speeds are strong enough in the model to initiate particle mobilization, while negative $u_{*ex}$

values imply the wind friction speeds are too weak. Immediately, we see that practically the entire domain is capable of lofting

dust in the $u_*$-driven simulations if sufficient dust and saltator material are present (Fig. 9a-d). Conversely, dust production in

the $u_{s*}$-driven simulations is restricted to isolated areas, including areas near the leading edges of individual storm cells (e.g.,

Fig. 9e-x). The $u_{*ex}$ results generated with globally scaled $U_{10m}$ and $C_s$ = 0.025 (Fig. 9e-h) are markedly similar to the $u_{*ex}$

values calculated with $U_{10m}$ scaled by $u_{ns*}$ (Fig. 9q-t). In particular, the excess friction speeds for these two instances match

rather closely over Arizona, with minor spatial extent differences for positive $u_{*ex}$ values in California and southern Nevada.

Thus, we chose $C_s$ = 0.025 as the $U_{10m}$ global scaling constant for our ALT4 test configuration. Interestingly, areas in the $u_{ns*}$

field equal to 0.025 (Fig. 6) tend to align with gridcells characterized as closed shrubland (Fig. 2) in the model domain where

there were strong winds, though this may be a coincidence.

  To further explore how our code modifications influenced simulated outcomes, we diagnosed the horizontal saltation flux for

the peak of the dust event associated with each test configuration (Fig. 10). Though the non-erodible area masking components

of the AFWA dust emission code are not applied in the AFWA module until the $F_B$ calculation (Eq. (6)), we included the

appropriate non-erodible area masks for the CTRL, ALT1, and ALT2 configurations (see Fig. 4) in the Fig. 10 $Q$ diagnostics.

Strong saltation occurred everywhere in the CTRL configuration plots unless the grid spaces were masked out (e.g., Fig 10a-

d). The four alternative model configurations showed similar $Q$ patterns to each other, suggesting that the masking had little

influence on the resultant $Q$ flux in module configurations that accounted for drag partition. The ALT1 and ALT2 $Q$ outcomes

were practically identical because the $z_0$ > 20 cm mask largely coincides with the areas where $S$ = 0 for this domain (e.g., Fig.

4). With the exception of a small area of saltation to the east of Phoenix, the ALT3 $Q$ results are also similar to ALT1 and

ALT2, which makes sense as developed/urban areas and higher-elevation forested regions generally yield low values of $u_{ns*}$.

Lastly, the ALT4 $Q$ spatial patterns mimic ALT3. However, the magnitude of the ALT4 $Q$ flux in the areas outside of where the

main dust event occurred are lower in magnitude than the ALT1, ALT2, and ALT3 results.

### 3.3 Simulated dust conditions

We compared our simulation results against observed $PM_{10}$ concentrations (Fig. 11-13). Note that $PM_{10}$ monitoring stations in

the United States are generally situated around populated areas, so most of the $PM_{10}$ sites associated with our model domain

were tightly clustered around Phoenix (e.g., Fig. 12b). The dust wall crossed over the Phoenix metropolitan area between 0000

and 0400 UTC on 4 July 2014. Importantly, the air quality conditions rapidly recovered once the main dust wall passed, and the

areas outside of the immediate dust event were relatively clear. According to surface weather station visibility, present weather

records, and observer remarks for Arizona and southern California (not pictured), it is unlikely the strong dust obscurations

extended much beyond the immediate convective cell outflow boundary area.





The original CTRL configuration produced a strong dust signal along the convective outflow boundary, but it also generated widespread dust concentrations in areas that in reality were clear (e.g., Fig. 11). Similar to the Q flux patterns, simulated $PM_{10}$

concentrations produced by the ALT1 and ALT2 configurations were nearly identical because dust in the $u_{s*}$-driven simulations did not originate near a masked area (see Fig. 4). Due to this artifact, we opted to limit our $PM_{10}$ assessment to the CTRL, ALT1, ALT3, and ALT4 simulations. The ALT1 configuration produced much lower $PM_{10}$ concentrations than the other test settings, with some dust emerging near the gust front. The ALT1 "dust wall," however, was an order of magnitude lower than the observed conditions. In contrast, the magnitude and spatial pattern of the simulated $PM_{10}$ values associated with ALT3 and

ALT4 aligned better with reported observations.

Though the general $PM_{10}$ patterns of the main dust event are similar between ALT3 and ALT4, the areas of maximum $PM_{10}$ extend further south along the gust front in the ALT3 simulation than in the ALT4 version (Fig. 11). We also note higher $PM_{10}$ values for ALT3 in southern California over the Mojave Desert and near the northeastern corner of Arizona compared to ALT4. However, the maximum $PM_{10}$ values associated with these two areas occurred in observation-limited regions. $PM_{10}$

observations from an EPA station in Barstow, California (Site ID: 06-071-0001; 34.8939080° N 117.024804° W) near the simulated area of maximum $PM_{10}$ over the Mojave Desert suggest that simulated $PM_{10}$ values in both ALT3 and ALT4 were higher than observed at this location (Fig. 13c). For example, simulated $PM_{10}$ values for ALT3 and ALT4 at the Barstow site peaked at 570 and 346 $\mu g$ m$^{-3}$, respectively, at 0000 UTC on 4 July 2014, while observed $PM_{10}$ values ranged between 5 to 47 $\mu g$ m$^{-3}$ during the 4 July 2014, 0000-0400 UTC period. However, the $PM_{10}$ concentrations in the simulations changed rapidly

over short distances near the Barstow EPA station as the simulations evolved. Thus, small shifts in the simulated dust position could greatly affect the apparent skill of the simulated output.

Surface wind speed observations from Edwards Air Force Base (KEDW, K9L2), which is primarily upstream and west of the simulated Mojave Desert $PM_{10}$ plume, and Barstow (KDAG) suggest that there were strong wind speeds over the area in question (Fig. 13). Observed wind speeds at the Barstow ASOS station closest to the simulated $PM_{10}$ plume ranged from ~5 to

10 m s$^{-1}$ between 0000-0300 UTC on 4 July 2014, which align relatively well with the modeled 10 m wind speeds. Though the simulated wind speed ramp up for the Barstow location was slightly early on 3 July 2014, the observed strong winds persisted longer over a similar time span before decaying. Accordingly, we can conclude that the model reproduced a realistic wind evolution pattern over the Mojave Desert. While the actual dust conditions for the area are less clear, the lack of commentary about dust in the recorded ASOS observer remarks leads us to believe that the simulated $PM_{10}$ values for both the ALT3 and

ALT4 configurations in the Mojave Desert region were higher than what occurred. However, the cause of this discrepancy is unclear. The erroneous $PM_{10}$ feature could be due to issues with the drag partition treatment, unresolved sediment supply influences, a combination of the two factors, or some other source of error within the model.

Though we are unable to do point-specific quantitative evaluations for the Phoenix metropolitan area due to the shifted location of the main dust-producing outflow boundary in the simulations, we can see from our spatial $PM_{10}$ analysis (e.g., Fig.

11) that the maximum $PM_{10}$ value during the peak of the dust event was generally co-located with the storm outflow boundary. Thus, we compared the simulated maximum $PM_{10}$ values for the area surrounding Phoenix with the maximum $PM_{10}$ values observed by the EPA stations as the dust wall passed over the city. Figure 12 shows timeseries of observed and simulated



maximum $PM_{10}$ concentration for the combined Maricopa and Pinal counties in Arizona. This combined county area generally encompasses the footprint of both the simulated and observed track of the storm. Note, the gust front in the simulation advanced

into the combined county area earlier than the recorded event (see Fig. 7), and the main dust wall moved away from the EPA stations around Phoenix after 0300 UTC. Therefore, we expect the simulated combined county maximum $PM_{10}$ values to be higher than the observed $PM_{10}$ conditions before and after the observed $PM_{10}$ peak at 0200 UTC on 4 July 2014. During the 0000 UTC to 0300 UTC period, the observed maximum $PM_{10}$ was was 9,132 $\mu$g m$^{-3}$, while the CTRL, ALT1, ALT3, and ALT4 configurations produced maximum $PM_{10}$ values for the combined county area of 17,151 $\mu$g m$^{-3}$, 419 $\mu$g m$^{-3}$, 8,539 $\mu$g

m$^{-3}$, and 7,240 $\mu$g m$^{-3}$, respectively. If we emphasize the order of magnitude over the exact value of $PM_{10}$ from these results to account for observed and simulated gust front location differences, both ALT3 and ALT4 produced reasonable results.

### 4    Discussion

Findings from this case study show that incorporating a drag partition treatment into the AFWA dust emission module can improve simulated dust transport for vegetated dryland regions. The CTRL simulation results with no drag partitioning align

with previous findings that the default AFWA module settings produce excessive dust conditions in the southwest United States. While the albedo-based drag partition method improved the results with respect to $PM_{10}$ spatial patterns and concentration magnitudes, we also achieved similar outcomes with a global wind scaling parameter.

We argue that the ALT4 globally scaled wind speed approach, at least how it's applied in our case study, should also be considered a crude form of drag partitioning. However, unlike the Chappell and Webb (2016) method, this global drag

partitioning approach assumes that the effects of the variability in vegetation cover over the dust-producing areas within the model domain are relatively negligible to the dust entrainment process. Our case study results imply that the spatial variability of $u_{ns*}$ was not a key factor in the evolution of the dust storm simulation, but this outcome may simply be an artifact of the conditions associated with our chosen case study. This particular dust event occurred in a confined, localized area due to its convective nature over a shrub-dominated landscape with relatively homogenous topography (e.g., Fig. 1-2). The spatial

variability of $u_{ns*}$ in the broader region may have more influence on dust events generated by synoptically driven wind forcing conditions (i.e., horizontal length scales on the order of several 100 to 1000+ km). Furthermore, the ALT4 configuration does not provide a viable modeling solution for users interested in exploring the effects of land management or land cover change on sediment transport and air quality and may be limited in its use to short term atmospheric forecasting applications.

It appears that drag partition treatments effectively suppressed the broadscale erroneous dust emissions in southern California

and southwestern Arizona inherent to the CTRL configuration without restricting the dust emissions that generated the Phoenix haboob. However, the relatively low $PM_{10}$ concentrations produced by the ALT1 simulation suggest that the $S$ parameter was masking the absence of drag representation in the original AFWA code. Although the model performance for this case study is arguably better with the drag partition when $S = 1$, this may not be the case for all regions. Further investigation on the role of the $S$ parameter in areas with more heterogeneous sediment supply is necessary to resolve this issue.





In order to better isolate the role of the $u_{ns*}$ spatial heterogeneity in our results from the effects of soil moisture on the dust emission process, we reproduced the excess friction speed diagnostics from Fig. 9 but used the threshold friction speed for air-dry soil conditions for saltation size bin 7 instead of the moisture-corrected threshold value ($u_{*ex\_dry}$; Fig. 14). The resultant $u_{*ex\_dry}$ values for all wind forcing treatments are, at least to some degree, largely higher in magnitude than their moisture-corrected counterparts in Fig. 9. However, both the $u_{*ex}$ and $u_{*ex\_dry}$ values for the forested areas along the Mogollon Rim

are generally low, even when using values for $C_s$ that align with the albedo-derived $u_{ns*}$ estimates associated with the barren and shrubland areas. For example, the $u_{ns*}$ values for the forested Mogollon Rim region in Fig. 6 are primarily less than 0.02 throughout the peak of the dust event. This suggests model sensitivity for this domain to the albedo-based drag partition may vary for alternate wind flow conditions and that the albedo-based drag partition scheme could potentially benefit from additional validation studies. However, modeled 10 m wind speeds will, in most cases, be reduced over forested areas due to the effects of

gridcell roughness on simulated wind speeds (aerodynamic roughness with respect to settings in the parent WRF model, not the drag partition treatment). Even if the $u_{ns*}$ values were high for a gridcell with forest cover, the simulated winds would likely be reduced by the internal model physics, potentially mitigating (or masking) a shortfall in the drag partition correction. These peculiar results over the forested area also highlight the need for consistency of aerodynamic roughness treatments between the dust emission module and other components of the parent weather model, including the land surface and planetary boundary

layer schemes, so that $u_*$ and $u_{s*}$ diagnosed for the dust emission treatment are consistent with the other model elements governing simulated wind flow. Findings by Webb et al. (2014b) demonstrate how these kinds of inconsistencies can lead to erroneously modeled increases in sediment transport with increasing vegetation coverage. As dust emission models like AFWA are further refined, careful consideration should be taken when configuring model components that incorporate land surface conditions on aerodynamic processes.

Even though we were able to largely replicate the $PM_{10}$ case study simulations produced with the albedo-based drag partition by using a global scaling factor, it is important to remember that our $C_s$ value selection was informed by the albedo-derived $u_{ns*}$ values and that the relatively short duration of our case study event meant that the $u_{ns*}$ values associated with our study were static throughout the event lifecycle. In any event, choosing an appropriate $C_s$ value *a priori* could be challenging when using the globally scaled surface wind speed approach for operational forecasting or climatological applications or for a larger

domain with more variability in vegetation coverage over dust-producing areas.

Spatial $u_{ns*}$ patterns derived from MODIS data will likely evolve over longer time spans from processes like annual vegetation growth cycles, drought conditions, anomalous wet periods, and land cover change. To investigate, we reviewed the temporal coefficient of variation (CV) of 500 m gridded monthly mean snow-masked $u_{ns*}$ values in our case study area over the 2001-2021 MODIS record (Fig. 15) and found that CV values generally ranged between 1-12% for grid pixels in the dust-

producing regions of southwest Arizona, with a few clusters of pixels ranging between 15-20% over active agricultural areas. By comparison, with the global $C_s$ drag partitioning (ALT4), we saw markedly different spatial patterns in areas of the model domain capable of initiating saltation simply by varying the $C_s$ parameter by increments of 0.005 (e.g., Fig. 9 and 14), or $C_s = 0.025 \pm \sim20\%$. Given that $u_{ns*}$ and $C_s$ are mathematically equivalent with respect to the model implementation (i.e., in terms of being multiplied by $U_{10m}$ to generate $u_{s*}$ in Eqs. 1 and 14, respectively), our MODIS-derived $u_{ns*}$ CV analysis suggests



that $C_s$ = 0.025 may be a reasonable setting for this domain in general. Nonetheless, though these CV values may appear to indicate a relatively low amount of variability in MODIS-derived $u_{ns*}$ over the majority of the dust-producing region, even modest changes in $u_{ns*}$ may result in simulated $u_{s*}$ values exceeding or falling below $u_{*ts}$.

Thus, the degree to which the observed variability in $u_{ns*}$ matters from a mesoscale dust modeling context is still unclear. Okin (2005) explored the influence of surface heterogeneity on saltation flux following the original method of Raupach et al.

(1993), where the drag partition adjusts $u_{*ts}$ directly rather than modifyng the $u_*$ value driving the saltation flux equation. When reviewing the daily saltation flux as a function of the CV of $u_{*ts}$, Okin (2005) found that CV values of 10% or less produced minimal differences in saltation flux. Additional modeling case studies focused on seasonal and interannual land cover changes are needed to determine the sensitivity of the AFWA dust emission model to $u_{ns*}$ variability.

Based on these findings, we argue that the Chappell and Webb (2016) drag partition approach may have the potential to

improve dust transport model performance in vegetated dryland regions. Still, it is worth discussing the nuances associated with its use. On the one hand, the technique removes the dependence on greenness-related remote-sensing indicators. By using albedo in a way that is interpreted as shadow to discern roughness elements and their associated aerodynamic properties, the Chappell and Webb (2016) technique may capture the drag induced by plants as well as drag generated by engineered and natural structures like fences, tillage furrows, and subgrid-scale topographic relief. In theory, this method incorporates

subgrid-scale heterogeneity and temporal variability in its calculation.

Another aspect to consider is the use of 10 m wind speed in Eq. (1), which is not physically based. The original derivation for $u_{s*}$ multiplied $u_{ns*}$ by a parameter Chappell and Webb (2016) referred to as "the freestream wind speed" or $U_f$ (also referred to by some studies as $U_h$). Yet this $U_f$ parameter is not associated with wind speed in the free atmosphere (i.e., wind speed above the atmospheric boundary layer). Instead, the value stems from the freestream flow associated with the wind tunnel data

collected by Marshall (1971) that Chappell and Webb (2016) used to develop their equations, which has no direct replacement. Additional research is needed to assess the degree of error introduced by the $U_f = U_{10m}$ assumption.

The Chappell and Webb (2016) scheme also assumes that all observed shadows emanate from solid subgrid-scale roughness objects within the flow field. This assumption could be obviated if shadows cast by tall objects extend into neighboring gridcells, creating a sensitivity to wind direction. While this is not likely to be an issue at the mesoscale or coarser modeling scales with

$u_{ns*}$ values derived from 500 m MODIS data, additional lower-bound grid resolution limitation studies may be necessary before applying the Chappell and Webb (2016) drag partition method to finer-scale dust transport models (e.g., large eddy simulation and field scale models) with $u_{ns*}$ initialized from high-fidelity data sources.

Like many other parameterizations, the Chappell and Webb (2016) drag partition is empirical. It incorporates multiple rounds of normalization and draws conclusions about nature from idealized "black-sky" (i.e., 100% direct illumination) scenarios that

do not exist in real-world settings. Furthermore, the Marshall (1971) wind tunnel experiments that Chappell and Webb (2016) used to establish their drag partition equations used non-porous, rigid cylinders to imitate natural roughness elements. Multiple studies have since demonstrated that the cylinder method is often a poor representation the aerodynamic behavior of live vegetation (e.g., Walter et al., 2012). Arguably, the effects of uncertainties in the rescaling parameters associated with the albedo approach on simulated dust entrainment outcomes may be comparable to errors created by uncertainties in inputs and





configuration parameters required for other drag partition methods. Additional studies comparing the many available drag
partitioning approaches in mesoscale and global scale dust transport models over extended periods of vegetation green-up and
scenecence are needed to fully resolve this issue.

Given the relative simplicity of implementing the Chappell and Webb (2016) drag partition into dust emission modeling
code, it would be worthwhile to continue investigating use of the scheme in convection-permitting and regional-scale dust
transport models. At these scales, many of the aforementioned issues would likely be small or negligible compared to errors
introduced by other parts of the modeling framework that govern $u_*$ and erodible sediment supply. In fact, some U.S. agencies
have already begun to incorporate the Chappell and Webb (2016) drag partition into their operational air quality models (e.g.,
Tong et al., 2020).

Finally, though the MODIS data preprocessing steps required by the Chappell and Webb (2016) drag partition are compara-
ble to the processing requirements of other air quality and weather model input datasets, operational environmental prediction
agencies may find the additional processing burden prohibitive. The computational resource allocations of production sys-
tems are often operating near capacity in order to meet a multitude of simulation parameter output requirements and data
dissemination schedules, making it challenging to justify added data ingest and preprocessing steps for a single model param-
eter. Accordingly, future efforts could explore the feasibility of using monthly or seasonal climatologies of $u_{ns*}$ for weather
forecasting applications.

## 5   Conclusions

This study explored the use of an albedo-based drag partitioning scheme introduced by Chappell and Webb (2016) to incor-
porate roughness effects into a widely-used dust emission and transport model within WRF-Chem. Our results for a convec-
tive dust event case study for the desert region around Phoenix, Arizona support previous findings that the original WRF-
Chem/GOCART/AFWA model configuration is prone to producing widespread erroneous dust emissions in the southwestern
United States. Furthermore, our findings suggest that incorporating a drag partition treatment into the AFWA dust emission
module may improve dust transport simulation and forecasting in vegetated dryland regions through better sheltering-effect
representation on dust emissions. This code adaptation could take the form of a formal drag partition treatment or a wind speed-
tuning approach if the roughness element coverage of the dust-producing areas associated with the model domain is somewhat
homogenous and the simulation is run over a short enough period that the surface roughness configuration does not change. We
note, however, that we cannot generalize model performance from a single case study event. Instead, we view this study as a
preliminary demonstration of capability and motivation to further explore the use of drag partitioning in dust transport models
to account for roughness effects on the dust entrainment process. Additional case studies and seasonal analyses are necessary
to determine if the Chappell and Webb (2016) drag partition method can reliably improve dust simulations used for various
weather prediction, land management, and climate change modeling applications.



*Code availability.* WRF-Chem v4.1 baseline source code and run instructions are available for download at https://github.com/wrf-model/ WRF/releases/tag/v4.1 (last access: 1 June 2020). The modified version of WRF-Chem v4.1 used to produce the results described in this paper is available for download via Zenodo (https://doi.org/10.5281/zenodo.6792554, Letcher et al., 2022). This modified source code includes additional runtime-activated functions that allow users to easily switch between the original AFWA dust emission module settings (i.e.,

CTRL) and the alternate configurations with drag partition included (i.e., ALT1, ALT2, ALT3, and ALT4). The Zenodo repository also includes a copy of the WRF Pre-processing System v4.2 code used to prepare the terrain and forcing data for our case study simulations. Detailed modified code descriptions, guidance for preparing the MODIS $u_{ns*}$ input data, and model runtime instructions are thoroughly documented in the report by Michaels et al. (2022).

## Appendix A:  Variable list

Tables A1-A2 provide the symbol, name, and value or description of variables referred to throughout this paper. Values of prescribed constants are listed. Variable arrays are described as "variable" for prescribed size bin-related settings, "spatially varying parameter" for static fields, and "spatiotemporally varying parameter" for temporally dynamic fields.

*Author contributions.* SLLeG developed the project concept and experimental design, performed the majority of the data analysis and interpretation, served as the primary manuscript author, and led project efforts. SD provided the MODIS-derived $u_{ns*}$ input datasets. TWL

modified the WRF-Chem code to implement the albedo drag partition into the AFWA module and produced the WRF-Chem case study simulation data. ARG, SLLeG, TSH, and TWL conducted the gust front location offset assessment. SD performed the $u_{ns*}$ CV analysis. GSO, NWP, and SLLeG collaboratively revised analyses and interpretation. All co-authors critically reviewed the manuscript.

*Competing interests.* The authors declare that they have no conflict of interest.

*Acknowledgements.* Funding support for this research was provided by the U.S. Army Engineer Research and Development Center (ERDC)

Extreme Terrain Research Program, "Forecasting EO/IR Extinction Characteristics for Active and Passive Optical Systems" project under Work Item H407HL, Funding Account Number U4365035 sponsored by the Assistant Secretary of the Army for Acquisition, Logistics, and Technology (ASA-ALT). NPW and SD were supported by NASA ROSES grant no. 80NSSC20K1673. Contributions by GSO were supported by NIH grant no. R01AI148336. Any use of trade, product, or firm names is for descriptive purposes only and does not imply endorsement by the U.S. Government. Permission to publish was granted by the ERDC Geospatial Research Laboratory. The USDA is an

equal opportunity provider and employer.



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

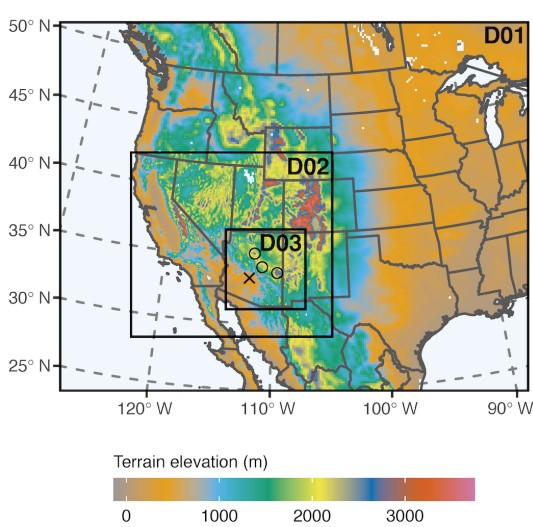

**Figure 1.** Telescoping model domains with shading showing terrain elevation in meters. The black × marks the location of Phoenix, Arizona. The elevated terrain feature northeast of Phoenix marked by a line of hollow circles is the Mogollon Rim.



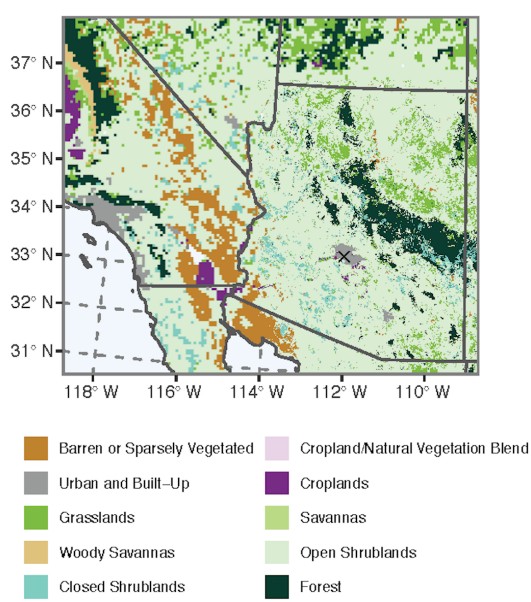

**Figure 2.** Prescribed land use categories assumed for the model domain. The black × marks the location of Phoenix, Arizona.



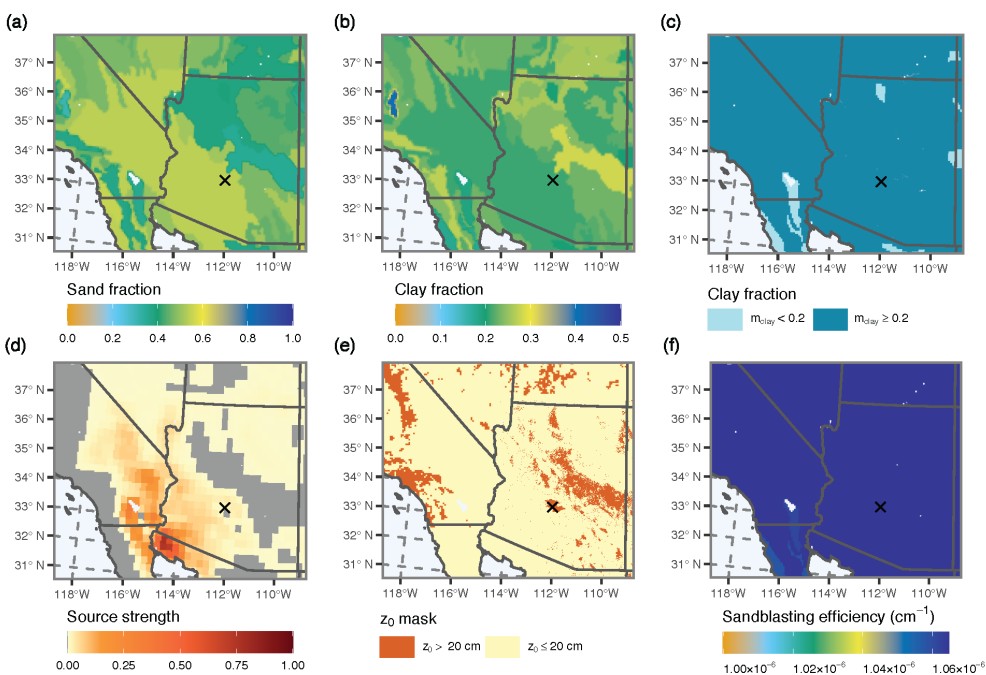

**Figure 3.** Important terrain attribute features considered in the AFWA dust emission flux calculation, including (a) soil sand fraction, (b) soil clay mass fraction ($m_{clay}$), (c) the clay mass fraction threshold affecting dust emission, (d) dust source strength ($S$; grey areas are masked), (e) the threshold aerodynamic roughness length ($z_0$) required for dust emission, and (f) sandblasting efficiency ($\beta$). The black × marks the location of Phoenix, Arizona.



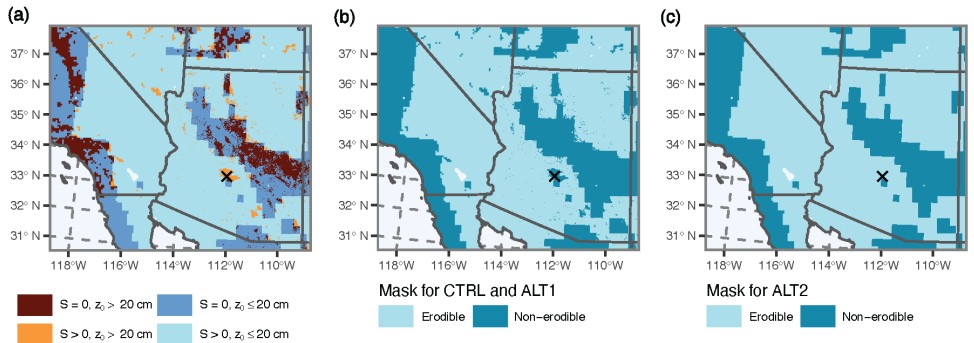

**Figure 4.** Erodible and non-erodible areas associated with the prescribed roughness and $S$ parameter masks. Erodible area masks in the plot on the left (a) largely overlap, where maroon indicates regions masked by both the $z_0$ conditional and $S$ parameter, the darker blue represents the area masked by only the $S$ parameter, and orange implies the masking is isolated to the $z_0$ conditional. Light blue regions in (a) have no masking. The middle plot (b) shows the combined $z_0$ and $S$ parameter masks applied in the CTRL and ALT1 configurations, and plot (c) shows the $S$ parameter mask applied in ALT2. No masks are applied in the ALT3 or ALT4 configurations. The black × marks the location of Phoenix, Arizona.





**Figure 5.** Schematic of the original and modified components of the dust emission module and their required inputs. A black diamond indicates the parameter varies spatially and temporally. A black circle indicates the parameter varies spatially but not temporarily, and a hollow diamond means the term is related to a particle size bin attribute table. The hollow circle indicates the field is derived from MODIS data. These input parameters are provided to the AFWA dust emission module by the WRF-Chem model. Color boxes imply which module components are used in the five different test configurations. See the comprehensive variable list in Table A1 for variable definitions.



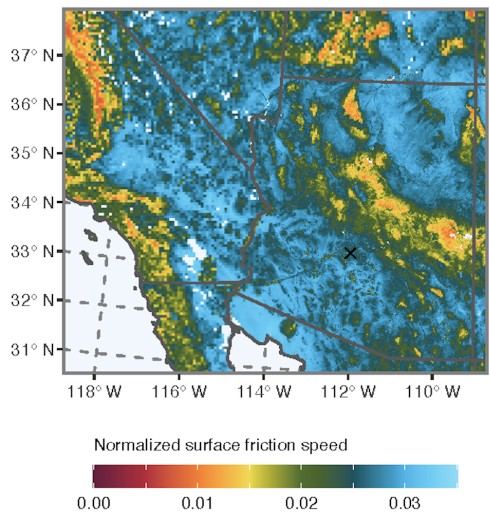

**Figure 6.** Normalized surface friction speed ($u_{ns*}$) diagnosed from MODIS data. Due to the relatively short duration of our case study event, the $u_{ns*}$ field remains fixed throughout the simulation. The black $\times$ marks the location of Phoenix, Arizona.



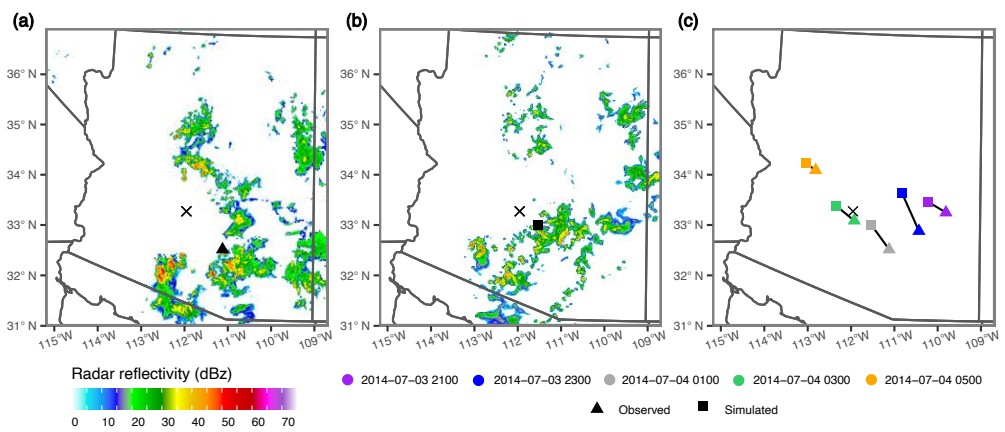

**Figure 7.** Observed and simulated radar reflectivity for 4 July 2014, 0100 UTC, including an (a) observed composite radar image and (b) the model-simulated radar image. The black × marks the location of Phoenix, Arizona. Triangle and square markers indicate the location of a central point along the leading edge of the convective cell that affected Phoenix in the observed and simulated storms, respectively. The plot on the right (c) shows how these leading edge central point locations evolved over time. Marker colors denote time in UTC.



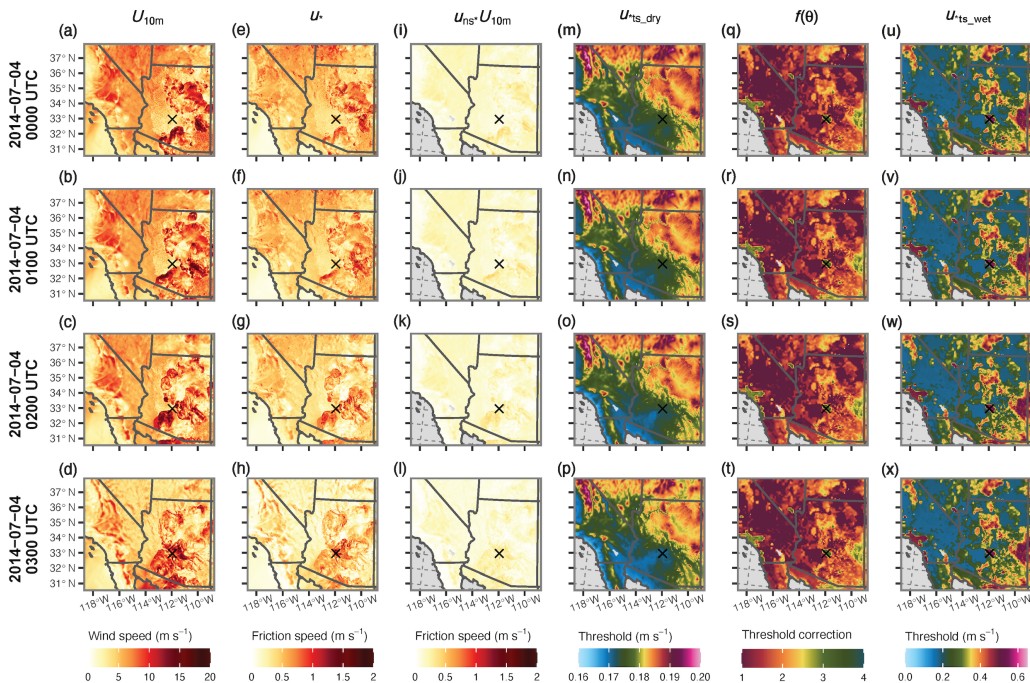

**Figure 8.** Simulated (a)-(d) 10 m wind speed ($U_{10m}$; m s$^{-1}$), (e)-(h) total wind friction speed ($u_*$; m s$^{-1}$), (i)-(l) surface friction speed determined from the albedo-based drag parameterization ($u_{s*}$; m s$^{-1}$), (m)-(p) threshold friction speed for a 69 $\mu$m diameter particle for an idealized smooth, dry, and barren soil surface ($u_{*ts}(D_{s,p} = 69\ \mu\text{m})$; m s$^{-1}$), (q)-(t) the soil moisture-based correction function ($f(\theta)$; dimensionless), (u)-(x) and soil wetness-corrected threshold friction speed for a 69 $\mu$m diameter particle ($u_{*ts}(D_{s,p} = 69\ \mu\text{m}, \theta)$; m s$^{-1}$) from 0000 to 0300 UTC on 4 July 2014. The $u_{*ts\_\text{dry}}$ and $u_{*ts\_\text{wet}}$ symbols used in the figure labeling represent $u_{*ts}(D_{s,p} = 69\ \mu\text{m})$ and ($u_{*ts}(D_{s,p} = 69\ \mu\text{m}, \theta)$), respectively. The wetness-corrected threshold is the product of the threshold friction speed and the soil moisture-based correction function. Note the differences in color bar scaling. The black $\times$ marks the location of Phoenix, Arizona.



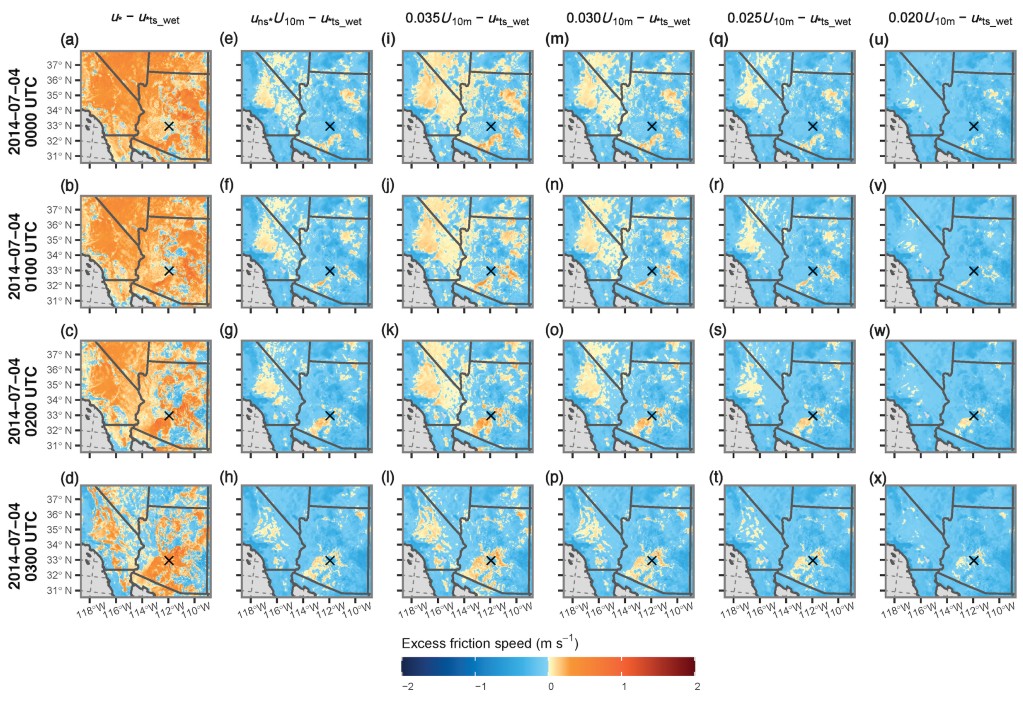

**Figure 9.** Excess friction speed ($u_{*ex}$; m s$^{-1}$) calculated by subtracting the moisture-corrected threshold friction speed for a 69 $\mu$m diameter particle, $u_{*ts}(D_{s,p} = 69\ \mu\text{m}, \theta)$, from the wind forcing parameter driving dust production from 0000 to 0300 UTC on 4 July 2014. The $u_{*ts\_wet}$ symbol used in the figure labeling represents $u_{*ts}(D_{s,p} = 69\ \mu\text{m}, \theta)$. Plots provided include (a)-(d) $u_{*ex}$ calculated using total friction speed, (e)-(h) surface friction speed determined from the albedo-based drag parameterization, and (i)-(x) four instances of 10 m wind speeds scaled using a global tuning constant as the wind forcing parameter. The black × marks the location of Phoenix, Arizona.

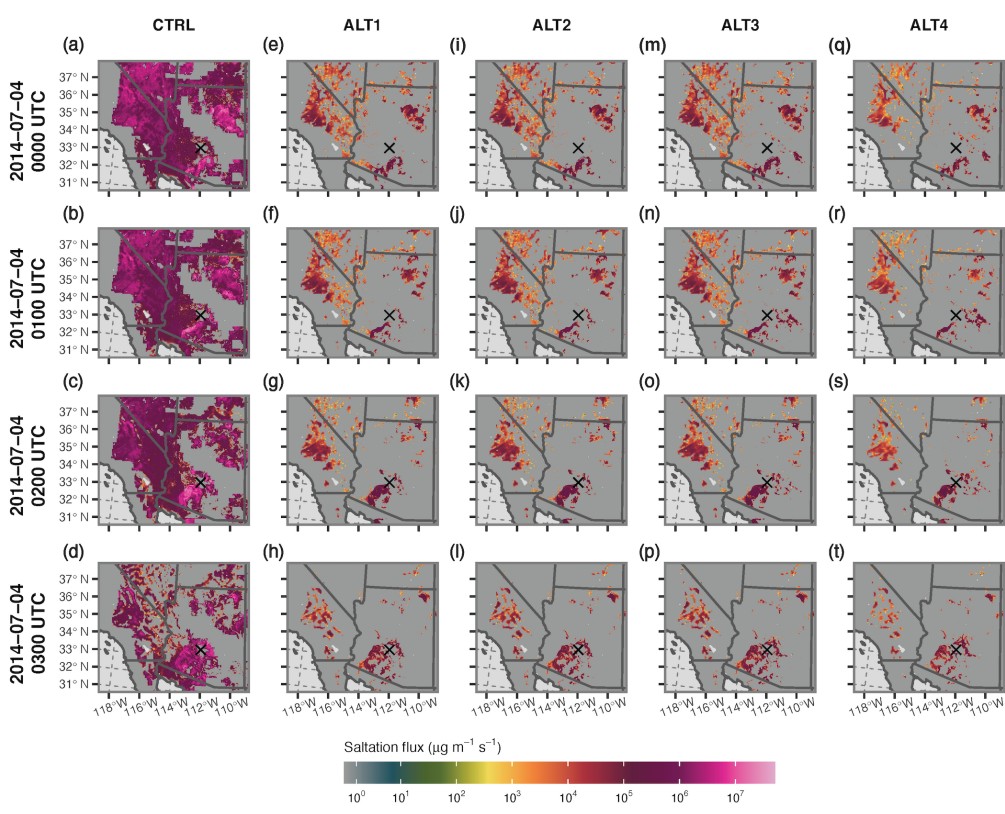

**Figure 10.** Horizontal saltation flux ($Q$; $\mu$g m$^{-1}$ s$^{-1}$) for each test configuration (CTRL, ALT1, ALT2, ALT3, and ALT4) with non-erodible area masking applied from 0000 to 0300 UTC on 4 July 2014. The black × marks the location of Phoenix, Arizona.

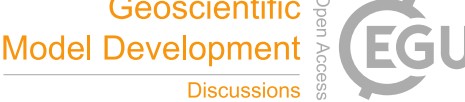

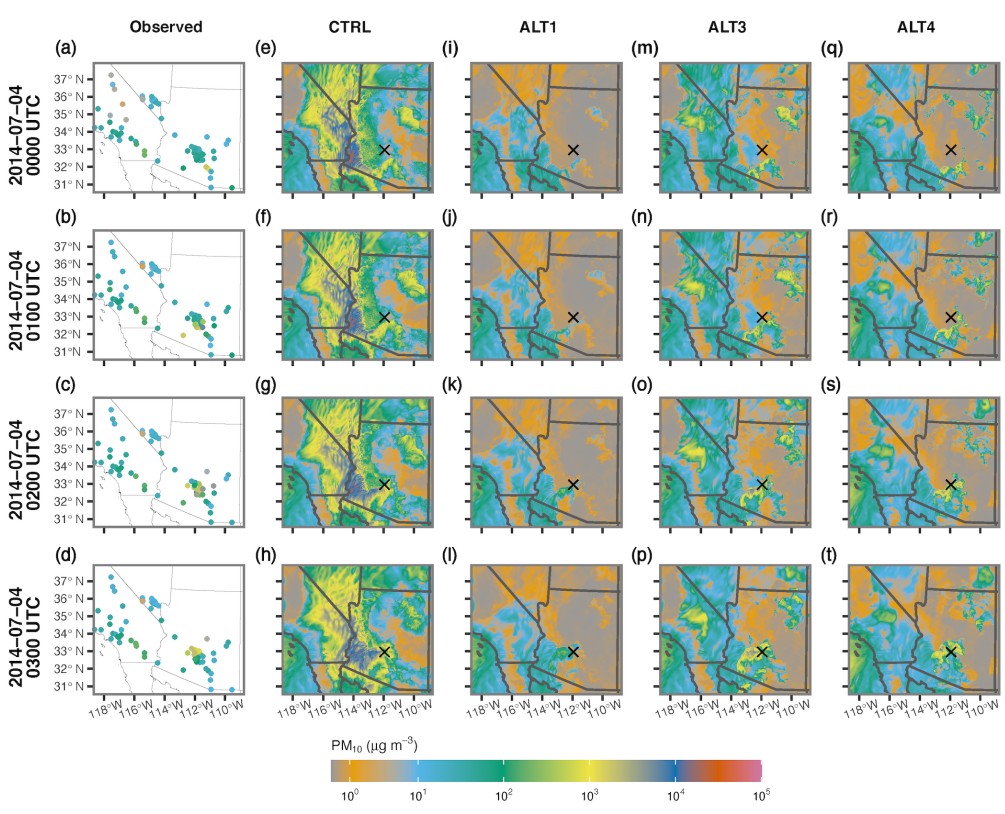

**Figure 11.** Observed and simulated $PM_{10}$ concentration for each test configuration (CTRL, ALT1, ALT3, and ALT4) in $\mu g$ m-3 from 0000 to 0300 UTC on 4 July 2014. The black × marks the location of Phoenix, Arizona. The ALT2 configuration results are not included in this figure since they are nearly identical to results from the ALT1 configuration.

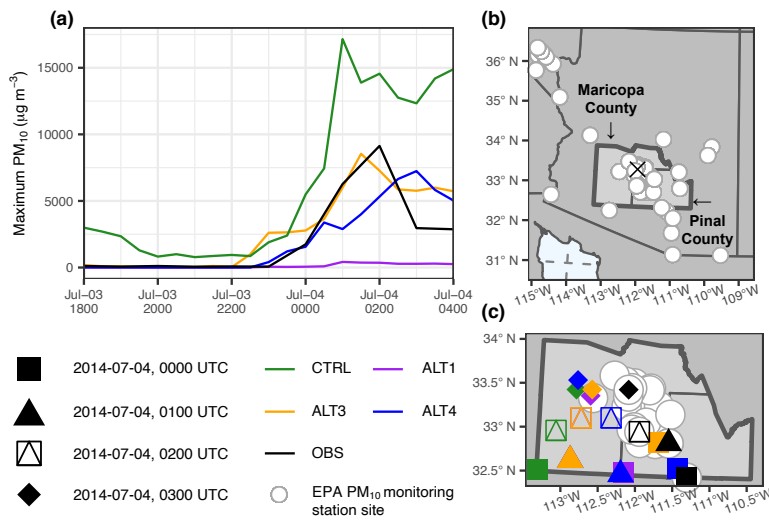

**Figure 12.** (a) Time series of observed (OBS) and simulated (CTRL, ALT1, ALT3, and ALT4) maximum PM$_{10}$ concentration in $\mu$g m$^{-3}$ for the combined Maricopa and Pinal counties in Arizona. Plot (b) highlights how most EPA monitoring stations in Arizona are in Maricopa County and Pinal County, clustered around the Phoenix metropolitan area (marked by the black ×). We used the hourly maximum PM$_{10}$ concentrations from the combined Maricopa and Pinal county areas for our assessment due to the shifted nature of the main dust-producing outflow boundary recorded by the EPA stations situated around Phoenix, Arizona in the simulations. The map zoom in (c) of the Maricopa County and Pinal County boundaries shows the EPA station locations relative to the hourly PM10 maximum values sampled for each test configuration and the observations between 4 July 2014, 0000-0300 UTC. The ALT2 configuration results are not included in this figure since they are nearly identical to results from the ALT1 configuration. Note the CTRL, ALT1, and ALT3 PM$_{10}$ maximum value sites overlap at 2014-07-04, 0100 UTC, and the ALT1 and ALT3 sites are co-located at 2014-07-04, 0200 UTC.



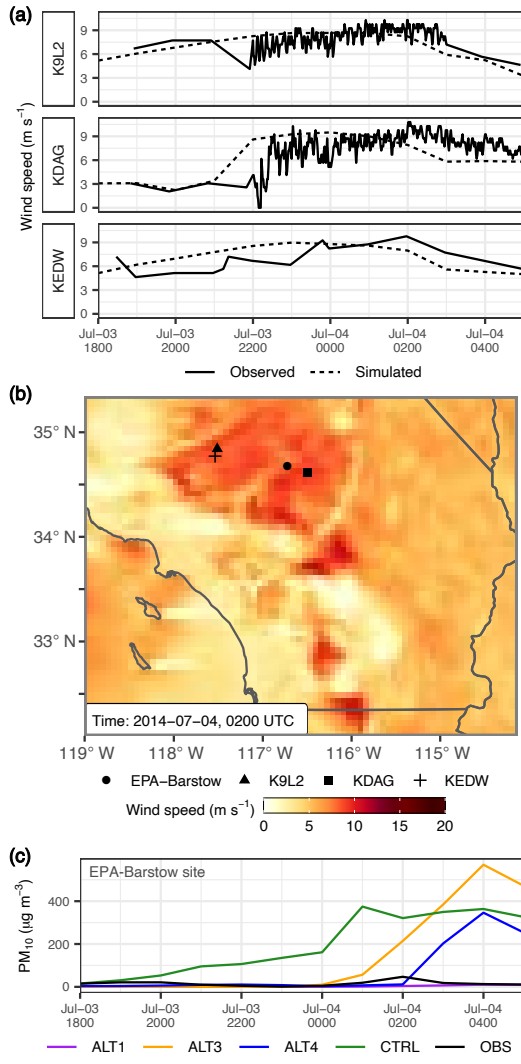

**Figure 13.** Simulated and observed (a) surface wind speed and (c) $PM_{10}$ values for ASOS and EPA station locations in the Mojave Desert area associated with the erroneously simulated $PM_{10}$ plume. The spatial plot (b) shows the station locations relative to the area of maximum wind speed simulated by the model over the Mojave Desert at 4 July 2014, 0200 UTC, when observed $PM_{10}$. values for the general location peaked.



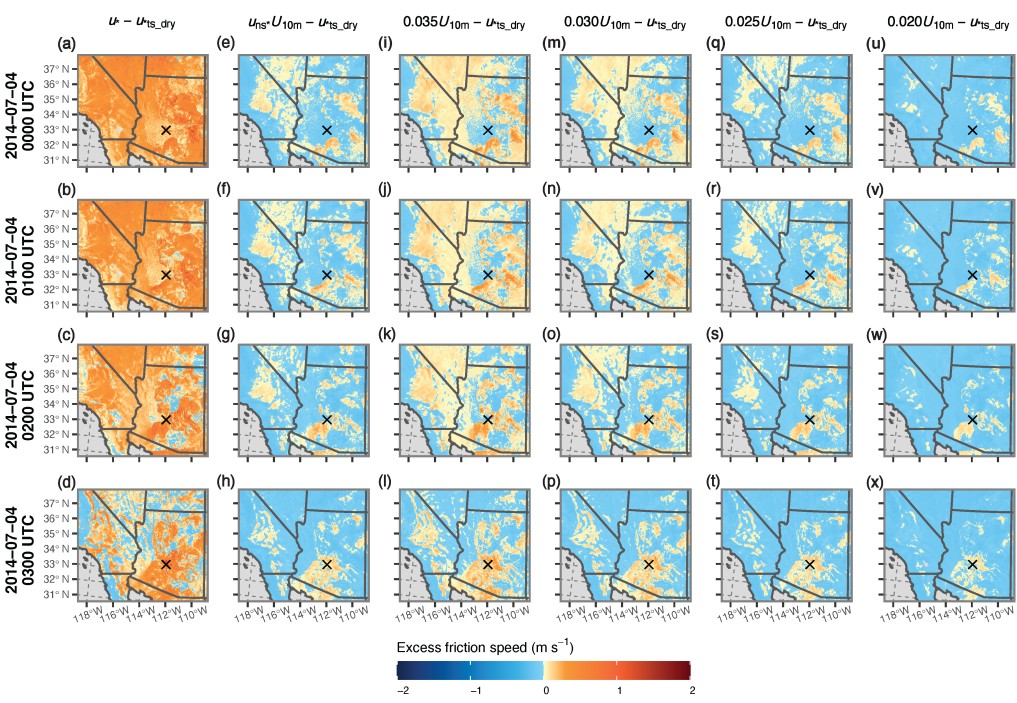

**Figure 14.** Excess friction speed ($u_{*ex}$; m s$^{-1}$) calculated by subtracting the moisture-corrected threshold friction speed for a 69 $\mu$m diameter particle, $u_{*ts}(D_{s,p} = 69\ \mu$m, $\theta)$, from the wind forcing parameter driving dust production from 0000 to 0300 UTC on 4 July 2014. The $u_{*ts\_wet}$ symbol used in the figure labeling represents $u_{*ts}(D_{s,p} = 69\ \mu$m, $\theta)$. Plots provided include (a)-(d) $u_{*ex}$ calculated using total friction speed, (e)-(h) surface friction speed determined from the albedo-based drag parameterization, and (i)-(x) four instances of 10 m wind speeds scaled using a global tuning constant as the wind forcing parameter. The black $\times$ marks the location of Phoenix, Arizona. Excess friction speed ($u_{*ex\_dry}$; m s$^{-1}$) calculated by subtracting the dry threshold friction speed for a 69 $\mu$m diameter particle, $u_{*ts}(D_{s,p} = 69\ \mu$m), from the wind forcing parameter driving dust production from 0000 to 0300 UTC on 4 July 2014. The $u_{*ex\_dry}$ symbol used in the figure labeling represents $u_{*ts}(D_{s,p} = 69\ \mu$m). Plots provided include (a)-(d) excess friction speed calculated using total friction speed, (e)-(h) surface friction speed determined from the albedo-based drag parameterization, and (i)-(x) four instances of 10 m wind speeds scaled using a global tuning constant as the wind forcing parameter. The black $\times$ marks the location of Phoenix, Arizona.





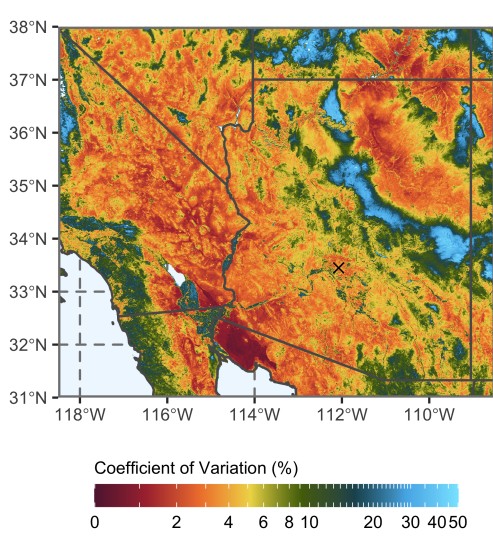

**Figure 15.** The spatiotemporal variability of $u_{ns*}$ for the 2001-2021 MODIS record shown as the coefficient of variation of monthly $u_{ns*}$ as a percentage. The black × marks the location of Phoenix, Arizona. Note the colorbar tick marks increase in increments of 5% after the 30% mark.





**Table 1.** Saltation particle size bins and their associated effective diameters (adapted from LeGrand et al., 2019, Table 1). Particle diameters are presented here in micrometers but handled in units of centimeters within the model.

| Saltation size bin | Effective diameter ($D_{s,p}$) ($\mu$m) |
| --- | --- |
| 1 | 1.42 |
| 2 | 2.74 |
| 3 | 5.26 |
| 4 | 10 |
| 5 | 19 |
| 6 | 36.2 |
| 7 | 69 |
| 8 | 131 |
| 9 | 250 |





**Table 2.** Dust particle size bins and their associated attributes (adapted from LeGrand et al., 2019, Table 2). Particle diameters are presented here in micrometers but handled in units of centimeters within the model.

| Dust size bin | Effective diameter ($D_{d,p}$) ($\mu$m) | Distribution fraction ($\kappa_{d,p}$) |
|---|---|---|
| 1 | 1.46 | 0.1074 |
| 2 | 2.8 | 0.1012 |
| 3 | 4.8 | 0.2078 |
| 4 | 9 | 0.4817 |
| 5 | 16 | 0.1019 |





**Table 3.** WRF-Chem physics and chemistry parameterizations. Note, the option numbers may be different in later model versions.

| Parameterization | Scheme | Namelist variable | Option |
|---|---|---|---|
| Cumulus | NSAS | cu_physics | 14 |
| (D01 & D02 only) | (Han and Pan, 2011) | | |
| Surface model | Noah | sf_surface_physics | 2 |
| | (Ek et al., 2003) | | |
| Surface layer | MYNN | sf_sfclay_physics | 5 |
| | (Nakanishi and Niino, 2004) | | |
| Boundary layer | MYNN 2.5 | bl_pbl_phsyics | 5 |
| | (Nakanishi and Niino, 2004) | | |
| Radiation | RRTMG | ra_sw(lw)_physics | 4 |
| (shortwave & longwave) | (Iacono et al., 2008) | | |
| Microphysics | Thompson | mp_physics | 8 |
| | (Thompson et al., 2008) | | |
| Chemistry | GOCART simple | chem_opt | 301 |
| | no ozone chemistry | | |
| Dust emission | AFWA | dust_opt | 3 |
| | (LeGrand et al., 2019) | | |
| Background emissions | GOCART simple | emiss_opt | 5 |
| Aerosol radiative feedbacks | Off | aer_ra_feedback | 0 |
| Aerosol optics | Maxwell approximation | aer_op_opt | 2 |





**Table 4.** Size-resolved saltation and bulk vertical dust flux treatments associated with each test case.

| Test | Size-resolved saltation treatment $Q(D_{s,p})$ | $u_{s*}$ equation | Bulk vertical dust flux treatment $(F_B)$ |
|---|---|---|---|
| CTRL | $\begin{cases} C\frac{\rho_a}{g}u_*^3\left(1+\frac{u_{*ts}(D_{s,p},\theta)}{u_*}\right)\left(1-\frac{u_{*ts}(D_{s,p},\theta)^2}{u_*^2}\right), & u_* > u_{*ts}(D_{s,p},\theta) \\ 0, & u_* \le u_{*ts}(D_{s,p},\theta) \end{cases}$ | | $\begin{cases} QS\beta, & z_0 \le 20 \text{ cm} \\ 0, & z_0 > 20 \text{ cm} \end{cases}$ |
| ALT1 | $\begin{cases} C\frac{\rho_a}{g}u_{s*}^3\left(1+\frac{u_{*ts}(D_{s,p},\theta)}{u_{s*}}\right)\left(1-\frac{u_{*ts}(D_{s,p},\theta)^2}{u_{s*}^2}\right), & u_{s*} > u_{*ts}(D_{s,p},\theta) \\ 0, & u_{s*} \le u_{*ts}(D_{s,p},\theta) \end{cases}$ | $u_{ns*}U_{10\text{m}}$ | $\begin{cases} QS\beta, & z_0 \le 20 \text{ cm} \\ 0, & z_0 > 20 \text{ cm} \end{cases}$ |
| ALT2 | $\begin{cases} C\frac{\rho_a}{g}u_{s*}^3\left(1+\frac{u_{*ts}(D_{s,p},\theta)}{u_{s*}}\right)\left(1-\frac{u_{*ts}(D_{s,p},\theta)^2}{u_{s*}^2}\right), & u_{s*} > u_{*ts}(D_{s,p},\theta) \\ 0, & u_{s*} \le u_{*ts}(D_{s,p},\theta) \end{cases}$ | $u_{ns*}U_{10\text{m}}$ | $QS\beta$ |
| ALT3 | $\begin{cases} C\frac{\rho_a}{g}u_{s*}^3\left(1+\frac{u_{*ts}(D_{s,p},\theta)}{u_{s*}}\right)\left(1-\frac{u_{*ts}(D_{s,p},\theta)^2}{u_{s*}^2}\right), & u_{s*} > u_{*ts}(D_{s,p},\theta) \\ 0, & u_{s*} \le u_{*ts}(D_{s,p},\theta) \end{cases}$ | $u_{ns*}U_{10\text{m}}$ | $Q\beta$ |
| ALT4 | $\begin{cases} C\frac{\rho_a}{g}u_{s*}^3\left(1+\frac{u_{*ts}(D_{s,p},\theta)}{u_{s*}}\right)\left(1-\frac{u_{*ts}(D_{s,p},\theta)^2}{u_{s*}^2}\right), & u_{s*} > u_{*ts}(D_{s,p},\theta) \\ 0, & u_{s*} \le u_{*ts}(D_{s,p},\theta) \end{cases}$ | $C_s U_{10\text{m}}$ | $Q\beta$ |





**Table A1.** Variable list.

| Variable | Name | Value |
|---|---|---|
| $a$ | Dimensionless scaling parameter | 0.0001 |
| $a_{mb}$ | Dimensional constant | 1331 cm$^{-x}$ |
| $b$ | Dimensionless scaling parameter | 0.1 |
| $b_{mb}$ | Dimensionless constant | 0.38 |
| $C$ | Dimensionless constant | 1.0 |
| $C_s$ | Global scaling factor | User determined variable |
| $c_v$ | Dimensional constant | $12.62 \times 10^{-4}$ cm |
| $D_{d,p}$ | Particle diameter of dust size bin $p$ | Variable |
| $D_{d,p\_max}$ | Maximum particle diameter of dust size bin $p$ | Variable |
| $D_{d,p\_min}$ | Minimum particle diameter of dust size bin $p$ | Variable |
| $\bar{D}_m$ | Emitted dust particle mass median diameter | $3.4 \times 10^{-4}$ cm |
| $D_{s,p}$ | Particle diameter of saltation size bin $p$ | Variable |
| $d\_\_$ | Distributions of particle property $\_\_$ | Variable field |
| $dM(s_{\mathrm{frac},s,p}, m_{\mathrm{clay}}, m_{\mathrm{sand}})$ | Particle mass distribution fraction for saltation size bin $p$ | Spatially varying parameter |
| $dS_{\mathrm{SFC}}(D_{s,p})$ | Basal surface coverage fraction for saltation size bin $p$ | Spatially varying parameter |
| $dS_{\mathrm{rel}}(D_{s,p})$ | Relative weighting factor for saltation size bin $p$ | Spatially varying parameter |
| $dV_{d,p}$ | Normalized volume distribution for dust size bin $p$ | Spatially varying parameter |
| $F_B$ | Bulk dust emission flux | Spatiotemporally varying parameter |
| $F(D_{d,p})$ | Dust emission flux for dust size bin $p$ | Spatiotemporally varying parameter |
| $f(\theta)$ | Wind friction speed threshold moisture correction function | Spatiotemporally varying parameter |
| $f_{\mathrm{iso}}$ | BRDF isotropic weighting parameter | Spatiotemporally varying parameter |
| $g$ | Gravitational acceleration constant | 981 cm s$^{-2}$ |
| $m_{\mathrm{clay}}$ | Soil clay mass fraction | Spatially varying parameter |
| $m_{\mathrm{sand}}$ | Soil sand mass fraction | Spatially varying parameter |
| $N_{\mathrm{SFC}}$ | Total basal surface area of soil bed | Spatially varying parameter |
| $N_V$ | Total normalized emitted dust volume | Variable |
| $Q$ | Horizontal saltation flux | Spatiotemporally varying parameter |
| $Q(D_{s,p})$ | Horizontal saltation flux for saltation size bin $p$ | Spatiotemporally varying parameter |
| $S$ | Dust source strength function | Spatially varying parameter |
| $s_{\mathrm{frac},s,p}$ | Soil separate class mass fraction for saltation size bin $p$ | Variable |
| $U_{10\mathrm{m}}$ | Wind speed at 10 m above ground level | Spatiotemporally varying parameter |
| $U_f$ | Freestream wind speed | Spatiotemporally varying parameter |
| $u_*$ | Wind friction speed | Spatiotemporally varying parameter |
| $u_{*ex}$ | Excess friction speed | Spatiotemporally varying parameter |
| $u_{*ex\_\mathrm{dry}}$ | Excess friction speed assuming air-dry soil conditions | Spatiotemporally varying parameter |





**Table A2.** Variable list continued.

| Variable | Name | Value |
|---|---|---|
| $u_{*ts}(D_{s,p})$ | Idealized smooth surface wind friction speed threshold required to mobilize particles associated with saltation size bin $p$ | Spatiotemporally varying parameter |
| $u_{*ts}(D_{s,p},\theta)$ | Idealized smooth surface wind friction speed threshold required to mobilize particles associated with saltation size bin $p$ corrected for soil moisture | Spatiotemporally varying parameter |
| $u_{ns*}$ | Soil surface component of wind friction speed normalized by the freestream wind speed | Spatiotemporally varying parameter |
| $u_{r*}$ | Roughness component of wind friction speed | Spatiotemporally varying parameter |
| $u_{s*}$ | Soil surface component of wind friction speed | Spatiotemporally varying parameter |
| $x$ | dimensionless constant | 1.56 |
| $z_0$ | Aerodynamic roughness length | Spatiotemporally varying parameter |
| $z_i$ | Terrain elevation above sea level of gridcell $i$ | Spatially varying parameter |
| $z_{\mathrm{max}}$ | Maximum terrain elevation above sea level in a $10° \times 10°$ area | Spatially varying parameter |
| $z_{\mathrm{min}}$ | Minimum terrain elevation above sea level in a $10° \times 10°$ area | Spatially varying parameter |
| $\beta$ | Sandblasting efficiency factor | Spatially varying parameter |
| $\theta$ | Soil moisture | Spatiotemporally varying parameter |
| $\theta_g$ | Gravimetric soil moisture | Spatiotemporally varying parameter |
| $\theta_{g'}$ | Gravimetric soil moisture without effect on capillary forces | Spatially varying parameter |
| $\theta_v$ | Volumetric soil moisture | Spatiotemporally varying parameter |
| $\kappa_{d,p}$ | Size distribution weighting factor for dust size bin $p$ | Variable |
| $\lambda$ | Crack propagation length | $12.0 \times 10^{-4}$ cm |
| $\phi$ | Soil porosity | Spatially varying parameter |
| $\rho_a$ | Air density | Spatiotemporally varying parameter |
| $\rho_{s,p}$ | Particle density of saltation size bin $p$ | Variable |
| $\rho_w$ | Water density | 1.0 g cm$^{-3}$ |
| $\sigma_s$ | Geometric standard deviation | 3.0 |
| $\omega$ | Albedo | Spatiotemporally varying parameter |
| $\omega_{\mathrm{dir}}(0°)$ | Black-sky albedo at nadir | Spatiotemporally varying parameter |
| $\omega_n$ | Normalized proportion of shadow (represented using a normalized albedo) | Spatiotemporally varying parameter |
| $\omega_{ns}$ | Empirically-scaled proportion of shadow (represented using empirically-scaled albedo) | Spatiotemporally varying parameter |