# Peer review of "Application of a Satellite-Retrieved Sheltering Parameterization (v1.0) for Dust Event Simulation with WRF-Chem v4.1"

_Geoscientific Model Development, 2022_

## Author Response (AR2)

**Response to Reviewers**

**Please note that all lines, tables, and figures referenced in our responses refer to the numbering in the revised version of the manuscript, unless otherwise specified.

**Reviewer #1:**

The manuscript entitled “Application of a Satellite-Retrieved Sheltering Parameterization (v1.0) for Dust Event Simulation with WRF-Chem v4.1” presents an albedo-based sheltering parameterization development to be used in dust transport modeling, namely WRF-Chem. The work presents a novel concept, and can potentially advance desert dust modeling. The structure of the paper is good, with extended information, clear methodology and solid scientific work. There is one major issue in my opinion that more testing should have been done for the domain configuration and there is a substantial lack of evaluation metrics. More information is given in the comments below.

Response: Thank you for your review. We have made several of the reviewer's suggested changes in the revised manuscript and believe it has strengthened the paper. Regarding the domain and model configuration comments, we did an extensive series of sensitivity tests to establish our model configuration and limit the potential for simulation errors created by issues with the environmental forcing conditions from the parent WRF model on the dust simulation. These efforts are thoroughly documented in the report by Gallagher et al. (2022).

The Gallagher et al. study investigated the sensitivity of the simulated forcing conditions from the parent WRF model driving the dust simulation to model initialization (spin-up) time, initial atmospheric conditions, model resolution (both horizontal and vertical), planetary boundary layer scheme settings, land surface model settings, cloud microphysics settings, and cumulus scheme settings. While there are certainly more model configurations that could be tested, findings from the Gallagher et al. study helped us establish WRF model settings that effectively simulated the convective structure, evolution, surface winds, and general placement of the storm cell that generated the case study dust event discussed here in our dust modeling paper. We stress that the Gallagher et al. (2022) report is meant to complement this work and focuses on many of the atmospheric components relevant to the reviewer’s concerns. The scope of our study here is to address the dynamic land surface and dust entrainment aspects of dust storm simulation, not the underlying atmospheric forcing studied in our related work.

We’ve updated the text from the beginning of Sect. 2.3 (model configuration) to make this clearer (Lines 231-250):

===

WRF-Chem is a version of the Weather Research and Forecasting (WRF) model by Skamarock et al. (2019) with additional modules for atmospheric chemistry processes and feedbacks (Fast et al., 2006; Grell et al., 2005). Like WRF, WRF-Chem is a fully compressible finite difference model that simulates atmospheric motion on the Arakawa C-grid and incorporates a variety of parameterizations for simulating sub-grid atmospheric motion, cloud microphysics, radiation, and terrain processes. We used WRF-Chem v4.1 for our test case simulation with WRF parent

model configuration settings suggested by Gallagher et al. (2022) and chemistry settings from LeGrand et al. (2019) and Letcher and LeGrand (2018). The study by Gallagher et al. investigated the sensitivity of WRF-simulated atmospheric forcing conditions for the dust event studied here. In particular, they focused on the effects of model initialization (spin-up) time, initial atmospheric conditions, horizontal and vertical model resolutions, and several WRF physics package settings to determine the optimal model configuration that minimized environmental forcing condition errors on the dust simulation.

Table 3 provides a brief overview of the model chemistry and physics configuration. However, complete pre-processor and run-time configuration files referred to as the namelist.wps and namelist.input files, respectively) for this effort are available from the code repository by Letcher et al. (2022) and in the report by Michaels et al. (2022). The model vertical grid used the default spacing distribution with 40 levels following a stretched hybrid sigma-pressure vertical coordinate that favors higher resolution near the ground. We note that the text and Table 2 from Gallagher et al. (2022) erroneously list their model configuration as having 42 vertical levels instead of 40. However, we confirmed their study simulations were generated with the same vertical level configuration used for our assessment. Figure 2 displays the three telescoping model domains (D01, D02, and D03, hereafter) with grid resolutions of 18 km, 6 km, and 2 km, respectively. In Fig. 3-4, we show key terrain attributes associated with the AFWA dust emission functions for D02 and D03, which primarily encompass the southwest U.S. desert region.

===

We recognize that some GMD readers outside the U.S. may have difficulty accessing the Gallagher et al. (2022) report from its official host site. A copy of the report is also available on ResearchGate:

https://www.researchgate.net/profile/Sandra-Legrand-2/publication/362509421_Simulating_Environmental_Conditions_for_Southwest_United_States_Convective_Dust_Storms_Using_the_Weather_Research_and_Forecasting_Model_v41/links/62ed81660b37cc3447718b53/Simulating-Environmental-Conditions-for-Southwest-United-States-Convective-Dust-Storms-Using-the-Weather-Research-and-Forecasting-Model-v41.pdf

Regarding the quantitative metrics, we added additional commentary on quantitative surface wind speed assessments (please see our response to a later comment).

Very well structured introduction with adequate information of available parameterizations. In Line 63 please add the reference: Spyrou, C.; Solomos, S.; Bartsotas, N.S.; Douvis, K.C.; Nickovic, S. Development of a Dust Source Map for WRF-Chem Model Based on MODIS NDVI. Atmosphere, 2022, 13, 868. https://doi.org/10.3390/atmos13060868, which is an up-to-date use of NDVI in defining dust sources. In Line 77 please add the reference. Skamarock et al. (2019). This reference is written later on, but it is best to put it here, where is the first mention of WRF.

Response: Thank you for the suggestions. We have added these references accordingly.

Section 2.1.1. This section is unnecessary large and mostly a repetition of the AFWA processes already described in other works. I would suggest limiting this section to half a page by only keeping the equations that are mostly relevant to this work. For instance the S parameter equation and analysis is not needed. The sentence "Essentially, S is a spatially varying tuning parameter ranging from 0 to 1 that assumes erodible material accumulates in low points in the terrain." is enough.

Response: We appreciate the reviewer's comment and agree that shortening the previously published model component descriptions would make the paper more streamlined. However, while there are several other works documenting AFWA module equations and the processes they represent, several published sources also contain misinformation about how the AFWA module works. This lack of consistency in the literature is likely due to the eight-year gap between when the AFWA code became publicly available through the WRF-Chem framework in 2011 and when the original developers published the first in-depth overview of the AFWA module in LeGrand et al. (2019). Due to the poor documentation heritage associated with the AFWA code, we strongly feel that GMD readers will benefit from a comprehensive overview of the AFWA module components discussed in this paper, especially with respect to parameters like the source strength field ($S$) that we removed or modified as part of our experiment.

Section 2.1.2. Line 177. The process by which the daily MODIS-derived fields are incorporated is not clear. Are they a part of the WPS process or a module is created that reads and re-grids the MODIS files while the model is running? As is written I assume this happens during runtime. Can you expand a bit?

Response: For this study, we incorporated the MODIS-derived fields through an auxiliary channel while the WRF-Chem simulation was running. The report by Michaels et al. (2022) fully documents how WRF-Chem pipes these data from the auxiliary feed through the chem driver to the AFWA module. The Michaels et al. report also provides detailed step-by-step instructions and scripts for acquiring, processing, re-gridding, and ingesting the MODIS fields, which we note in this section and again in the model code availability section near the end of the manuscript. While it makes sense to eventually add these processes to the WRF Preprocessing System (WPS), these additional steps were beyond the scope of our study.

We've updated the sentence on Lines 189-191 to clarify:

To incorporate $u_{s*}$ into the AFWA dust emission module, we configured WRF-Chem to ingest daily MODIS-derived $u_{ns*}$ fields (Eq. (11)) that had been interpolated to the model grid domain into the WRF-Chem framework through an auxiliary channel at model run time and modified the dust emission equations to use $u_{s*}$ in place of $u_*$.

Section 2.1.2. Line 179. You use the 10m wind speed that is derived while WRF is running. Why not use the first model level wind speed? In general we try to avoid the 10m speed as the 10m wind components are diagnostic quantities. If we need wind speeds this close to the ground it is best to lower the first model level as close as we can and increase the vertical levels used. This is critical as the dust emissions are very sensitive to small changes in wind (as the authors state). If possible I would like to see changes between using 10m wind speed and first level wind speed

(where the first level is close to 10m). It is entirely possible that the differences are negligible and 10m wind speed is adequate.

Response: Thank you for this insightful comment. After looking into this, we found that our lowest model level is already approximately 10 m above ground level, and differences between the lowest model level wind and 10 m wind fields are relatively negligible. Finally, we ran a brief test simulation that used the first model level winds instead of the 10 m winds and found no major changes in the dust simulation; therefore, we think that in this case, and in cases with a sufficiently high vertical resolution near the surface, the 10 m wind speeds are appropriate.

However, there are additional considerations we would like to address. Specifically, we caution against using the lowest model level winds in place of the 10 m diagnostic wind speed here for a few reasons:

1. First and foremost, we're applying an established methodology from Ziegler et al. (2020) to explore if the albedo-based drag partition parameterization in its current form can improve AFWA module-simulated dust emission patterns. Critically, the Ziegler et al. approach derives the albedo-based partition using the 10 m wind speed as an input, so it is possible that wind speeds associated with heights closer to the land surface may worsen outcomes.
2. The empirical components of the $u_{ns*}$ equation (Eq. (11)) were initially derived relative to what Chappell and Webb (2016) referred to as the freestream wind speed flowing above engineered roughness elements in a wind tunnel environment. This freestream wind speed value may not have a direct physical equivalent to a real-world setting, but replacing it with wind speed values closer to the immediate ground surface would not make sense in this context.
3. The core AFWA module equations are based on wind friction speed, not wind speed. From the AFWA module perspective, changing the vertical model level heights will not directly affect the AFWA module calculations. Wind speed is only used in the conversion of $u_{ns*}$ to $u_{s*}$.
4. Our lowest model level being situated approximately 10 m above ground level is a coincidence. We used the WRF v4.1 default vertical atmospheric level distribution set by the WRF model's real executable (*real.exe*). Older or newer versions of WRF may not adhere to this standard. Furthermore, these particular vertical-level settings may not always be appropriate for all domain or case study forcing conditions. For example, the current WRF v4.1 hybrid vertical coordinate is not a consistent height above ground level. Instead, it is dependent on the vertical distribution of temperature and pressure, especially close to the ground. For events with dramatic changes in temperature and pressure, the effective height of your lowest model level can vary both in time and space, whereas the 10 m diagnostic wind ensures a common reference height. Setting a dependency on the vertical level configuration may make it challenging for others to apply the drag partition treatment in their respective WRF-Chem model configurations.

Section 2.2. Just a small note for those unfamiliar with the area, the dust source area should be noted clearly.

Response: Unfortunately, we cannot directly attribute dust emissions for this event to a specific source location because the satellite imagery was cloud-obscured. For this analysis, we can only speak to "dust sources" in terms of how the aerodynamic roughness length ($z_0$) and vegetation masks are applied to their respective model configurations [e.g., Fig. 5 (previously Fig. 4)]. It's important to remember that a general lack of widespread dust entrainment in our simulated test cases configured with a drag partition treatment does not necessarily imply a lack of dust sources in the associated area. Rather, roughness elements may have suppressed dust generation by blocking or reducing momentum transfer from the atmosphere to the soil surface. We've added an additional "storm summary" figure to Sect. 2.2 to help readers conceptualize the general placement and forcing conditions associated with the main dust wall to help alleviate confusion regarding where dust entrainment likely occurred.

[Figure]

**Figure 1.** A summary overview of the atmospheric forcing conditions associated with a convective dust storm that occurred 3-4 July 2014. The event was characterized by a persistent broad high pressure (blue H), clockwise upper-air circulation (black streamlines and arrows), mid- to low-level moisture transport (blue arrows) from the Gulf of California and the Gulf of Mexico, and surface wind vectors (purple arrows) converging downslope of the Mogollon Rim. These conditions initiated and sustained several convective systems that merged near and around Phoenix, Arizona (denoted with a black ×). The shaded overlay shows the national radar reflectivity composite imagery at the storm's peak intensity on 4 July 2014, 0100 UTC. Shortly after, the gust front at the leading edge of the convective cell south of Phoenix (dotted black line) moved northwest (storm motion denoted by thin black arrows) over the greater Phoenix area. A wall of thick dust associated with the storm lofted and transported along the gust front.

Section 2.2. The meteorological conditions and weather patterns that led to this event should be described in detail. For example Mean sea level pressure and wind patterns at the surface and at 850hPa should be added (even from the model simulations, if weather maps are not available). Is the event related to a density current? I see later on that you use NEXRAD. Is this the reason?

Response: The atmospheric evolution of this event is fully explored in the report by Gallagher et al (2022) referenced at the beginning of Sect. 2.2. Note, the Gallagher et al. report goes into great detail on the synoptic, mesoscale, and local conditions associated with the entire lifecycle of our focus dust case study event using a broad collection of analysis fields, radar composites, and observations for support. The new conceptual storm summary figure mentioned above should help readers visualize the general forcing conditions associated with the storm event.

We've added the following sentences to the end of the first paragraph of Sect 2.2 (Lines 219-223):

Figure 1 provides a conceptual overview of the general environmental forcing conditions associated with the dust event. For a more in-depth review of the storm evolution, including synoptic, mesoscale, and local condition assessments using a broad collection of analysis fields, radar composites, and observations for support, we encourage readers to review the Gallagher et al. (2022) report.

(Please note - the new Fig. 1 referenced here was not part of the original manuscript submission.)

Figure 2. Mark the X spot more clearly. Add a circle maybe?

Response: We've enhanced the Phoenix marker in Fig. 2 (now Fig. 3) to make it stand out better.

Section 2.3. Line 233. The 12 hour initialization is not adequate to generate a proper dust concentration background. In general 5-15 days are needed for this, but seeing as the dust event is very quick and localized one can assume that 12 hours is enough. Still this needs to be expanded upon.

Response: We agree with the reviewer that extended model spin-up times are often necessary for spinning up background dust (and other aerosols). Indeed, the majority of the dust associated with our case study event was localized and produced by dust lofting along a convective outflow boundary. The Gallagher et al. (2022) study reviewed the model sensitivity to initialization time and found that extending the spin-up time to 24 hours (i.e., starting the simulation on 2 July 2014, 1200 UTC instead of 3 July 2014, 0000 UTC) caused the simulation to diverge from the observed pre-convection environment, degrading the overall simulation accuracy. The aforementioned update to Sect. 2.3 (model configuration) notes this model spin-up time assessment.

Section 3. A more thorough statistical analysis is needed. There are no statistical indexes calculated. Also the text structure is a bit confusing. In my first read I thought that no timeseries was created until I saw figures 12 and 13. This needs to be written again in a more concise and

analytical way. A statistical evaluation should also be performed, even a rudimentary one with all the available data for wind speed and PM10. Unfortunately qualitative analysis in not enough.

Response: Thank you for your comments. It's not entirely clear which parts of the text's structure were confusing to the reviewer. However, we have attempted to improve the text. In particular, we added a few introductory sentences to the beginning of Sect. 3 to help shape the organization.

Lines 322-325: Experimental results are reviewed as follows. We begin with an overview of the $u_{ns*}$ field, then review the environmental forcing conditions and dynamic components of the dust emission scheme simulated by the model. Lastly, we assess the resultant dust-related parameters produced by each test configuration outlined in Table 4. This holistic component-based approach helps to break down how the many factors affecting modeled dust conditions contributed to each test simulation outcome.

We also updated sentences introducing Fig. 9-15 where appropriate. While Fig. 9-15 are all time series plots, Fig. 9-12 and Fig. 15 are spatial time series plots. To help clarify, we have added "time series of <parameter>" text lead-ins throughout Sect. 3-4, including figure captions.

The Gallagher et al. (2022) companion assessment included a statistical analysis of surface wind speeds in the innermost model domain where the main dust event occurred. They found that the average wind speed bias for the entire forecast period was +0.59 m $s^{-1}$. However, most of this overestimation occurred at night, outside the main convective period. We have updated the text from the first paragraph of Sect. 3.2 accordingly:

Lines 338-347: The model was able to reproduce the storm's general structure and timing, including the formation of the initial quasi-linear convective system and the collapse of the convective line into individual cells (results consistent with findings by Gallagher et al., 2022). Furthermore, the simulated near-surface wind speeds were in good agreement with wind speeds observed at ASOS stations. However, simulated wind speeds peaked 1 to 2 hours early in some locations with slightly higher (about +1 m $s^{-1}$) intensity. According to Gallagher et al., these minor wind speed errors may be partly due to erroneous land use characterization, particularly in the higher terrain elevation areas where the storm initiated. Gallagher et al. also performed a full statistical analysis of simulated surface wind speeds against all available ASOS wind speed data from the innermost domain (D03). The average wind speed bias for the entire forecast period was +0.59 m $s^{-1}$. However, a large portion of this overestimation occurred during non-convective nocturnal periods (3 July, 0500–1500 UTC and 4 July, 0800–1600 UTC).

Statistical analyses of $PM_{10}$ are less straightforward. As discussed in Sect. 3.3, the EPA $PM_{10}$ stations are not equally distributed across the domain. Instead, these stations are tightly clustered around population centers (e.g., the dense station network surrounding the Phoenix metropolitan area). As a result, any misalignment in storm position can substantially affect the reliability of point-based $PM_{10}$ station comparisons. This is especially important to consider for our case study given the slight position and timing offset of the storm (e.g., Fig. 8c). Hence, we chose to limit our $PM_{10}$ assessment to a qualitative analysis of the maximum $PM_{10}$ value simulated along the gust front.

Some studies (e.g., Hyde et al., 2018) compare hourly average $PM_{10}$ observations against hourly average simulated $PM_{10}$ values on the county level. While this may make sense for widespread dusty conditions, this approach may not work well for highly localized haboob conditions like our focus case study event. We tested this hypothesis for our case study using the combined Maricopa/Pinal county area as our region of interest (e.g., Fig. 13), keeping in mind that the main dust wall crossed directly over most of the $PM_{10}$ stations surrounding Phoenix. This assessment approach made the CTRL configuration appear to perform better than the ALT3 and ALT4 configurations artificially because the two alternate configurations incorporated several grid cells with low $PM_{10}$ values (correctly) in areas with no $PM_{10}$ station coverage. If we attempt the same exercise with hourly maximum values for the combined county area (which largely mirror our outflow boundary max $PM_{10}$ assessment shown in Fig. 13) instead of the county-averaged hourly values, we still end up with deceptive results due to the minor position and timing offsets that affect when the simulated storm entered/exited the combined Maricopa/Pinal county area boundaries.

Accordingly, we maintain our position that point-based $PM_{10}$ quantitative analyses for this case study event would be misleading.

The authors also state that "small shifts in the simulated dust position could greatly affect the apparent skill of the simulated output". This is correct but an effort should be made to setup the model in such a way to try to see if the wind and dust forecasts can be improved. Even using different initial conditions, or SST. Right now the selection of the domain was done based on another work which provided good results, but maybe this setup is not adequate for this study. More testing is needed in order to have a proper domain basis to evaluate the methodology.

Response: Please see our previous comments and text adjustments about the parent WRF model configuration. We appreciate the reviewer's comment (and recognize the importance of correctly simulating the environmental forcing conditions for simulated dust entrainment assessments). However, we also acknowledge that the mesoscale details of convective system evolution are a source of irreducible uncertainty within WRF. For example, operational mesoscale models like the High-Resolution Rapid Refresh (HRRR; e.g., Benjamin et al., 2016), often experience difficulties with the timing, location, and morphology of convective storms. So, while the large-scale forcing conditions and convective initiation were well captured by our simulation, the exact timing, shape, and location of the resulting convective system were subject to error. Of the two challenges, we consider it more important for the simulation to capture the structure and underlying dynamics of the storm rather than the precise location, as the latter is easier to adjust and account for than a misrepresentation of the observed quasi-linear convective system (QLCS) as a cluster of thunderstorms, supercell, or mesoscale convective system. Additionally, there was an extensive amount of work put into determining our model configuration to limit and document errors in the predicted wind field in Gallagher et al. (2022).

The results section is clear and the shortcomings of the methodology are presented. I would like to see a more extensive analysis on the benefits of using the proposed methodology in dust modeling

Response: We appreciate the reviewer's comment. For this preliminary analysis, the benefits of the methodology are manifest in the vast improvements we see with ALT3 and ALT4 over the initial CTRL configuration. This paper aims to show the weakness of the existing approach, and in the context of a single storm, introduce the adapted module with a drag partition included. We agree that continued research is needed (which we highlight in both the discussion and conclusions). If future studies warrant, continued use of satellite-derived roughness information and its effects on dust emission in the AFWA module could markedly improve investigations of the role of short- and long-term changes in vegetation on dust emission patterns. This, in turn, could be of benefit to model users interested in drought hazard, climate change, land management, and post-wildfire condition modeling applications.

We added the following commentary to the end of the conclusion section (Lines 611-616):

The benefits of using a drag partition methodology in the AFWA module are manifest in the vast improvements we see with ALT3 and ALT4 over the initial CTRL configuration. Follow on studies are still necessary to explore if these benefits persist over longer simulation periods. However, we anticipate that satellite-derived roughness information could markedly improve investigations of the role of short- and long-term changes in vegetation on dust emission patterns. This, in turn, could be of benefit to model users interested in drought hazard, climate change, land management, land use/land cover change, and post-wild fire condition modeling applications.

Should the above be addressed I would like to see this work published in GMD.

Response: Thank you. We appreciate the support.

**Reviewer #2:**

This paper implements a vegetation sheltering parameterization into the WRF-Chem AFWA dust model and tests it for a case study in the American southwest. Previously, vegetation coverage was inferred from "greenness" factors, which may under-represent brown and non-photosynthetic vegetation in arid regions. The parametrization here uses vegetation shadows derived from MODIS to determine a vegetation height to represent roughness lengths.

Overall, the paper is well-written and straightforward and I would like to see it published in GMD. The manuscript has the potential to advance the representation of dust emissions in numerical models and constrain the scalable factors inherent to dust parameterizations. The motivation and application of the sheltering factor seems solid, but only a single case study (with mostly qualitative results) is presented to test the new parameter.

Thank you for your review. We have made several of the suggested changes in the revised manuscript and believe it has strengthened and clarified the paper.

Major Comments

1) The main finding (at least for this case study – taken from Hyde et al. 2018) is that switching to the sheltering parameter decreases the area of dust source regions and

therefore dust emissions. The case study was selected because the default AFWA scheme initially overpredicted dust, so implementing the sheltering parameter would naturally lead to a better fit between modeled and observed dust. However, there were an equal number of cases from the Hyde et al. (2018) ensemble that showed AFWA underestimating dust, which means the sheltering factor would lead to a worse fit. We can't infer the impact of this parameter from a single case. I suppose once the parameter is released to the community it will be tested more and time will tell if it ends up being used. But, it would go a long way to test this parameter for a case study where dust was underpredicted too.

Response: Thank you for this comment. While we acknowledge that broad conclusions cannot be extrapolated from a single case study review (which we also state in our conclusion section), we have chosen not to include additional case studies for the following reasons:

1. This paper is primarily meant to serve as an introduction to and discussion on the usage of the albedo-based drag partition in the WRF-Chem AFWA module, rather than a robust evaluation of the scheme. We fully agree with the reviewer that additional studies are necessary to achieve that goal and hope this paper will serve as the basis from which the broader modeling community can build on.

2. We wanted to maintain a focus on a well-studied and well-simulated convective event that occurred in an area with a robust $PM_{10}$ monitoring network.

We recognize that the Hyde et al. (2018) study concluded that the AFWA module tended to both over- and underestimate dust concentrations. However, we have some concerns about the Hyde et al. evaluation methodologies that lead us to suspect the underestimated cases may be an artifact of misleading data interpretations. Specifically, Hyde et al. compared hourly average $PM_{10}$ observations against hourly average simulated $PM_{10}$ values averaged over the county level. While this may make sense for widespread dusty conditions, this approach may not work well for highly localized haboob conditions, especially since the $PM_{10}$ monitoring stations are not evenly distributed in this domain (e.g., the $PM_{10}$ monitoring stations are primarily clustered around the Phoenix, Arizona metropolitan area). For example, the model may incorporate several grid cells with low $PM_{10}$ values (correctly) in areas with no $PM_{10}$ station coverage, causing the simulations to look artificially lower than the observed patterns. The reverse could also occur if the storm is offset from the sensor locations, making the simulated dust concentrations appear higher than they should be.

We also note that the Hyde et al. study primarily assessed the representativeness of their forcing conditions by comparing simulated rainfall accumulation patterns to radar data. While they supplemented this analysis with comparisons of maximum simulated and observed wind speed assessments, their general approach is not a sufficient means for properly evaluating the storm structure and morphology. This is especially important for convectively driven case studies, where storm morphology is equally important to the overall intensity of surface winds for effective dust event simulation. It's possible that some of the Hyde et al. case studies captured the rain patterns but not the outflow boundary conditions that actually drove the dust (e.g., intense convective storms with strong gust fronts versus poorly organized "squishy rain blobs"). Since the Hyde et al. paper only reviews model performance for one of their nine dust event case

studies, it is difficult to tell if all of their dust simulations are reliable assessments of the AFWA module.

That said, since we knew the dust event associated with our chosen case study passed directly over the Phoenix area monitoring stations, we were more likely to accept the Hyde et al. conclusion that the AFWA module grossly overestimated dust conditions. Furthermore, the AFWA module has been available to the WRF-Chem user community for over a decade. Anecdotally, community members have shared with us that they tend not to use the AFWA model for this domain without making substantial tuning or input dataset adjustments because of its propensity to overestimate dust production in the area north of the Gulf of California (which we clearly see in our CTRL simulation).

2) There is a lot of confusion and debate in the dust parameterization literature over "roughness" factors and the scaling of dust emissions based on vegetation. Partly it's because there are multiple roughness effects and the terminology gets muddled. Thus, some schemes are probably double counting the effects of surface roughness on dust emissions (Webb et al., 2020). I recommend overexplaining what this new shielding term is representing physically and to be more explicit in what all these roughness effects and dust source terms do (sections 1-2). I have added the prefix "Terms" to specific comments where terminology could be confused and more explanation would be helpful to readers. For instance, does it or other terms represent the production or dissipation of momentum by roughness elements (or is that a PBL scheme effect?)? Or just shielding (i.e. dust gets caught in an obstacle or canopy and can't loft freely)? What about plant (stem/trunk) area reducing bare soil area? Etc.

Response: We appreciate the reviewer's comment and acknowledge the legacy of terminology confusion in dust literature. We have attempted to clarify terminology throughout the paper accordingly. Please note, we are a bit hesitant to explicitly state what the albedo-based drag partition scheme represents physically due to some of the issues we raise in our discussion section. Rather, we prefer to think of it as a parameterization for representing a component of the dust emission process that the default AFWA dust emission module wouldn't be able to characterize otherwise. The albedo scheme is not sophisticated enough to differentiate obstacles blocking mobilized sediments from reductions in wind shear stress. From our perspective, it offers an empirical relationship between shadowing and aerodynamic parameters, that, in theory, can be used to estimate drag partitioning conditions and their general effects on dust entrainment.

We added additional commentary to the Eq. (10) description to help clarify how the normalized albedo calculation links to area-integrated roughness conditions.

Lines 181-183: In essence, Eq. (10) integrates the albedo across viewing angles for a single illumination angle (solar noon), producing an areal shadow estimate that, in theory, represents non-erodible roughness element conditions for an integrated (500 m pixel) area (e.g., Fig. 1 from Ziegler et al., 2020).

3) There is little discussion about the role of meteorology in the dust forecast. Since dust emissions scale as windspeed^3, small wind speed errors can lead to large dust errors. It's always hard in dust modeling to tell where the errors come from – the meteorology or the dust scheme. I would like to see more justification for why the errors in this case study were determined to be from the dust scheme and not the meteorology.

Response: We fully agree with the reviewer that thoroughly assessing the simulated environmental forcing conditions driving the dust event is an essential step in the dust scheme evaluation process. The Gallagher et al. (2022) paper referenced in both the case study description (Sect. 2.2) and the model configuration overview (Sect. 2.3) was meant as a complement to this study. These authors conducted an in-depth review of our case study event evolution and performed extensive model configuration sensitivity studies to determine "best recommendations" for our parent WRF model initialization source and model configuration settings. Gallagher et al. also evaluated and documented errors in simulated forcing conditions for our case study event so this paper could focus on addressing the dynamic land surface and dust entrainment aspects of dust storm simulation rather than the atmospheric components.

We agree that our initial manuscript submission did not make this clear and have updated the text accordingly. For example:

The first paragraph of Sect. 2.3 (model configuration); Lines 235-240: We used WRF-Chem v4.1 for our test case simulation with WRF parent model configuration settings suggested by Gallagher et al. (2022) and chemistry settings from LeGrand et al. (2019) and Letcher and LeGrand (2018). The study by Gallagher et al. investigated the sensitivity of WRF-simulated atmospheric forcing conditions for the dust event studied here. In particular, they focused on the effects of model initialization (spin-up) time, initial atmospheric conditions, horizontal and vertical model resolutions, and several WRF physics package settings to determine the optimal model configuration that minimized environmental forcing condition errors on the dust simulation.

The first paragraph of Sect. 3.2 (environmental forcing simulation results); Lines 338-347: The model was able to reproduce the storm's general structure and timing, including the formation of the initial quasi-linear convective system and the collapse of the convective line into individual cells (results consistent with findings by Gallagher et al., 2022). Furthermore, the simulated near-surface wind speeds were in good agreement with wind speeds observed at ASOS stations. However, simulated wind speeds peaked 1 to 2 hours early in some locations with slightly higher (about +1 m $s^{-1}$) intensity. According to Gallagher et al., these minor wind speed errors may be partly due to erroneous land use characterization, particularly in the higher terrain elevation areas where the storm initiated. Gallagher et al. also performed a full statistical analysis of simulated surface wind speeds against all available ASOS wind speed data from the innermost domain (D03). The average wind speed bias for the entire forecast period was +0.59 m $s^{-1}$. However, a large portion of this overestimation occurred during non-convective nocturnal periods (3 July, 0500–1500 UTC and 4 July, 0800–1600 UTC).

Please note that we do highlight issues with the simulated meteorology in our discussion on the rotation/position and timing of the gust front (e.g., Sect. 3.2; Paragraph 2 and Fig. 8). Due to

these meteorological discrepancies, we adjusted our evaluation approach and limited our simulated dust assessment to a qualitative evaluation of the overall storm system rather than performing a more traditional point-based comparison against hourly $PM_{10}$ observation records.

4) How much of the PM10 is from other aerosol species than dust in the model?

Response: We included the standard suite of aerosols covered by the GOCART "simple" module configuration option in WRF-Chem for completeness and to activate the model $PM_{10}$ calculation functions. This setting incorporates dust, sea salt, black carbon, organic carbon, and dimethyl sulfide (DMS) into the background $PM_{10}$ estimates. In this case, most $PM_{10}$ comes from either dust or sea salt transport. Sea salt and DMS distributions are relatively isolated to areas over the ocean and the immediate coastlines. Dust is the primary source of $PM_{10}$ over land. Contributions from black carbon, organic carbon, and DMS were negligible, except in dense urban areas along the southern California coastline. All inland $PM_{10}$ estimates on the order of 300 $\mu g\ m^{-3}$ or higher were primarily due to dust transport.

We added the following text to the beginning paragraph of Sect. 3.3; Lines 427-432:

To ensure our simulated $PM_{10}$ conditions could primarily be attributed to dust transport, we also reviewed the general contribution of each aerosol species to the overall simulated $PM_{10}$ patterns (not pictured). For this case, most of the simulated $PM_{10}$ came from either dust or sea salt transport. Sea salt and dimethyl sulfide distributions were relatively isolated to areas over the ocean and the immediate coastlines, while dust was the primary source of $PM_{10}$ over land. Contributions from black carbon, organic carbon, and dimethyl sulfide to simulated $PM_{10}$ totals were negligible, except in dense urban areas along the southern California coastline.

Specific Comments [Line Numbers or Section]

[44-46] – A lack of representation of roughness elements is one reason for poor dust forecasts, the way it's written here makes it seem like it's the only or the major reason. Other important reasons would be model resolution, representation of cold pool and precipitation processes, source grid map, etc. Bukowski & van den Heever have done some work on the role of dust-lofting cold pools and model resolution (2020), but they also have a new paper (2022) showing that surface type and roughness effects are the most sensitive / important factor for predicting dust concentrations in cold pool dust events (haboobs) – similar to the July 2014 case study modeled here. This reference may help with motivations for this paper.

Response: Thank you very much for pointing us toward this paper. We added the following sentence to the end of the final paragraph in our introduction section accordingly (Lines: 103-105):

This hypothesis is further strengthened by recent findings from Bukowski and van den Heever (2022), who found accurate roughness effect characterizations are critical for predicting dust patterns associated with cold pool events similar to our chosen case study.

Regarding challenges with model configurations mentioned by the reviewer, we agree that these are important elements to consider. However, we opted to focus our introduction on challenges specific to dust emission modeling that extend beyond issues common to atmospheric modeling in general.

[48] – How is U* calculated in the model? Is it diagnosed like U10? In Eq. 1 Us* is a function of U10 and not U* - just checking that the model level / physical processes going into these equations are the same for comparing CTRL and the ALT simulations.

Response: The WRF model estimates spatiotemporally varying values of $u_*$ in the surface layer scheme that handles critical parameters for simulating the vertical behavior of mean airflow and turbulence properties within the lower bounds (approximately the lowest 10%) of the atmospheric boundary layer. For our experimental configuration, this occurs in the MYNN surface layer module (*module_sf_mynn.F*). The dynamic 2-dimensional $u_*$ scalar value is not associated with a specific model level and gets derived through the scheme's similarity theory functions used to parameterize turbulent closure schemes for atmospheric conditions near the land surface. Our WRF configuration also calculates the $U_{10\text{m}}$ diagnostic in the same MYNN surface layer module (lines 1221-1224) using the $u_*$ parameter and a semi-empirical log-wind profile relationship for diagnosing wind speeds at different heights above ground level.

The reviewer is correct that the $u_{s*}$ parameter is a function of the model 10 m wind speed diagnostic (and MODIS-derived fields) and not $u_*$; however, we note that the parent WRF model uses $u_*$ to diagnose the 10 m wind speed diagnostic. As a result, $U_{10\text{m}}$ and $u_*$ should exhibit similarities in their general spatial patterns under stable atmospheric conditions.

Please see our response to the next comment for more information on how $u_*$ relates to $u_{s*}$.

[51] – The approach here is to modify the surface Us* to include roughness elements (surface and above). But with the drag partitioning method of splitting up U*, there is also an Ur* term to represent roughness effects. Why did the authors seek to modify the Us* to include roughness elements instead of incorporating the shielding term into Ur*?

Response: We focused on $u_{s*}$ instead of $u_{r*}$ since dust emits from the soil. The wind shear stress acting on the roughness elements ($\tau_r$) and the immediate soil surface ($\tau_s$) are related and must sum to the total wind shear stress ($\tau$) acting on the land surface. Note that $\tau_r \propto u_{r*}^2$, $\tau_s \propto u_{s*}^2$, and $\tau \propto u_*^2$. This conservation, therefore, means that the sheltering (shielding) term is already accounted for by $u_{s*}$.

We added a more thorough overview of the drag partitioning concept to the introduction section to help clarify this point and address similar topics brought up in later comments.

Lines 48-55: Sediment mobilization schemes are often represented in terms of wind friction speed, $u_*$, a scalar parameter commonly used to describe processes related to wind shear stress ($\tau$; note that $\tau = \rho(u_*)^2$, and $\rho$ is air density). Near the land surface, $u_*$ represents the total wind shear stress ($\tau$) acting on both the horizontal soil surface ($\tau_s$) and roughness elements ($\tau_r$) (i.e., $\tau =$

$\tau_s + \tau_r$; see Raupach, 1992; Raupach et al., 1993). This process is typically termed *drag partitioning* and is often expressed in terms of $u_*$ rather than $\tau$. Since $\tau$ is proportional to $u_*$, we can similarly divide $u_*$ into soil surface ($u_{s*}$) and roughness ($u_{r*}$) components (i.e., $u_* = u_{s*} + u_{r*}$). The wind shear stress that reaches the immediate soil surface (i.e., $\tau_s$) governs particle mobilization, so dust emission models driven by $u_{s*}$ (or wind erosivity) instead of $u_*$ may produce better outcomes (e.g., Darmenova et al., 2009; Okin, 2008; Webb et al., 2020).

[65] – What about roughness elements like biocrusts, which are typically flat and sprawling?

Response: While technically any landscape element (biocrusts included) contributing to atmospheric drag will, in turn, affect dust emission, we are unaware of any studies quantifying the biocrust contribution to the drag partition. In theory, the albedo-based drag partitioning scheme should pick up on any raised element casting a shadow. At present, we tend to focus more on larger roughness elements like trees, shrubs, grasses, topographic features, and man-made structures. However, we suspect that dust emission modeling uncertainties caused by poor representation of biocrust soil aggregate binding effects on sediment supply probably far outweigh soil crust-related drag partitioning simulation errors (e.g., Rodriguez-Caballero et al., 2022).

[94-95] – Terms: describe more what the drag partition here refers to (Ur*?)

Response: See previous response. We have updated the part of the introduction (Sect. 1) where we introduce drag partitioning to help clarify this term.

[95-96] – What about a dust underprediction event? See major comment #1

Response: We fully acknowledge the need for and encourage the future study of the proposed parameter for a broader variety of dust events. An underprediction case study would be a particularly interesting research case. With respect to the findings of Hyde et al. (2018), we refer back to our response with regard to the gaps and flaws in their methodology that lead to those conclusions. Specifically, the underlying atmospheric component is critical to resolve and evaluate correctly with respect to underestimating dust concentration. A simulated convective storm may reflect the maximum intensity and direction of observed wind speeds at the surface but have a smaller spatial footprint, completely different storm structure, or notable shift in geographic location that can suppress the resulting dust emission and transport. As such, our study refers frequently to the conclusions of Gallagher et al. (2022) to ensure the accuracy of the atmospheric forcing conditions and give us confidence that our results primarily reflect the nuances of the dust emission module. We highly encourage future and follow-on studies to apply a similarly critical eye to the underlying atmospheric state.

[98-100] – What about the meteorology? What if this convective case study is just difficult to get right?

Response: Please see our previous comments. We agree with the reviewer that accurately simulating the environmental forcing conditions is a critical step in the dust modeling process. We also agree that convectively driven case studies add an extra layer of complexity to the

problem. However, we took several steps in our modeling setup phase to limit the potential for errors (e.g., Gallagher et al., 2022) and account for them in our dust modeling assessment when they occurred.

[132] – Terms: is S the so-called "erodibility" map in some models?

Response: We note that some communities and published sources have labeled the source strength parameter (*S*) used in the AFWA module as an "erodibility" map. However, we prefer to avoid that description. Others have called it a "dust source" map, which is also inaccurate. Ginoux et al. (2001) initially established the *S* parameter as a means for integrating large-scale areas of loose erodible sediment supply primarily associated with Holocene-era mega lakebeds into a relatively coarse-resolution global dust transport model. It was a creative way of incorporating sediment supply into a dust transport modeling framework using readily available parameters with global coverage in the absence of real data.

Adaptations of the Ginoux et al. (2001) topographic dust source strength parameterization approach are used in several modeling frameworks (e.g., Barnum et al., 2004; Collins et al., 2011; LeGrand et al., 2019, Vukovic et al., 2014), which is not surprising given that it was one of the first globally portable modeling techniques established for characterizing dust sources, is relatively easy to implement, and is readily accessible as a pre-calculated field available for download from multiple sources. However, the term "erodibility map" has been applied to several terrain-based datasets over the years (e.g., Cremades et al., 2017; Ginoux et al., 2001; Grini et al., 2005; Jugder et al., 2018; Kimura, 2016; Parajuli et al., 2014; Zender et al., 2003). The phrase has become a sort of "catch-all" term for fields used to characterize some aspect of sediment supply or dust source strength. It's difficult to know which model or field the reviewer is referring to here.

We updated the *S* field description in Sect. 2.2.1 to the following (Lines 151-157):

The *S* parameter (originally described by Ginoux et al., 2001) represents the availability of loose erodible soil material at a given location based on the degree of topographic relief of the surrounding area. This approach assumes that soil composition remains consistent over time, and the simulated land surface will neither run out of dust material nor acquire new dust material through fluvial or atmospheric deposition as the simulation evolves. Some papers refer to this spatially varying *S* field as an erodibility or dust source map. However, both labels provide an inaccurate description of how *S* functions in the AFWA module. Accordingly, we will not use that terminology here. Essentially, *S* is a spatially varying tuning parameter ranging from 0 to 1 that assumes erodible material accumulates in low points in the terrain,...

[134] – Terms: aerodynamic roughness length – also is this part of the double-counting problem?

Response: Here, the aerodynamic roughness length ($z_0$) parameter refers to the length scale or height above ground level where the mean atmospheric flow, but not turbulent flow, is integrated to zero. The AFWA module uses a simple $z_0$ = 20 cm threshold to prevent dust emission from forested and urban areas (e.g., LeGrand et al., 2019). In this capacity, the $z_0$ mask has the

potential to create a "double counting" problem for vegetation once the new drag partition is added, which is why we remove it as part of our experimental design.

The ALT1 and ALT2 model configurations are nearly identical, with the only difference being that we removed the mask for $z_0$ > 20 cm from the ALT2 bulk vertical dust emission flux equation (Eq. (6); see Table 4). In this specific case study, however, removing the $z_0$ mask had little effect on the simulated dust patterns due to the overlap in the areas masked by $z_0$ and the vegetation mask built into the *S* parameter field (e.g., Fig. 5).

We've added the following text to the Eq. (6) description (Lines 144-147):

Here, $z_0$ represents the theoretical height at which the mean wind speed near the surface falls to zero due to surface-induced drag (e.g., Stull, 1988; Zobeck et al., 2003). Importantly, this $z_0$ parameter is not part of the new drag partition treatment, but rather a value from the parent WRF model used for a variety of land surface and air flow processes and diagnostics.

[143-151] – The description of S is confusing. Probably don't need the original formulation or Eq. 8, just how it is used here.

Response: Thank you for your comment. While the formulation of the *S* parameter has been covered by several other published sources, there are also several published sources containing misinformation on how the *S* term in the AFWA module is calculated and what it represents. Due to the poor documentation heritage associated with the AFWA module and the *S* term in general, we strongly feel that GMD readers will benefit from a comprehensive overview of how the S parameter constructs, especially because we remove this particular field as part of our experiment.

We've updated the section in an attempt to clarify the content from our perspective. In response, we've edited for clarity where we perceived areas for improvement (Lines 161-165):

Of note, the AFWA module uses interpolated values of *S* initially derived from a 1/4° elevation dataset. In addition, the *S* field incorporates a vegetation mask that blocks dust emission (i.e., *S* = 0) from vegetated areas derived from a 1° × 1° resolution land cover dataset. While these settings may be appropriate for some modeling applications, the coarse nature of these input datasets likely limits the spatial viability of *S* at mesoscale and convective-permitting model resolutions (e.g., Saleeby et al., 2019; Vukovic et al., 2014; Walker et al., 2009).

[154-155] – Walker et al. (2009) and Saleeby et al. (2019) are good references for showing the effect of high-resolution dust source maps for mesoscale modeling applications.

Response: Thank you for the suggestion. We've added these references to the text.

[169] – Terms: "normalized" appears in the name of Uns* (normalized surface friction speed) - is the "normalization" from the albedo normalization by Fiso or some other part? When I see "normalized...speed" my assumption that speed is the normalizing variable in the factor but I don't think that is the case here.

Response: From this comment, we understand that the reviewer is asking about different parameters by which something is normalized. Indeed, the normalization in $u_{ns*}$ is different from the normalization in $\omega_n$. We use the terminology and symbology for $u_{ns*}$ adopted from previously published resources. Further, we think that the equations are sufficiently clear regarding the use of $u_{ns*}$ in the model and what factors go into the normalization process.

[180-181] – Dust schemes are mostly based on empirical fits to data. Was Eq. 13 fitted to some data that might be affected by simply substituting Us* into it? I.e. if Eq. 13 was tuned to dust observations, changing the denominator might de-tune that relationship.

Response: Per LeGrand et al. (2019), $C$ is set to 1.0 in the AFWA module based on recommendations from Darmenova et al. (2009), Laurent et al. (2006), and Marticorena et al. (1997). We did not attempt to refit or tune the empirical constants associated with Eq. (13) or any other equation in the model for this study. Modifying tuning parameters in a meaningful way with a single case study event would be difficult at best. While tuning can be useful, it is generally best done after reviewing simulation outcomes for several events and extended periods. Even with tunings applied, the "best" parameter configuration may only be relative to a specific model configuration, domain setting, application, or region. Additional research would be necessary to discern if additional tuning adjustments are needed to fully optimize the dust model performance with a drag partition included.

[185 - 187] – Terms: excess wind friction speed. What is this physically? It seems like a model diagnostic more than something physical.

Response: Excess wind friction speed is the wind shear stress that governs mass flux. We have rephrased the text to help clarify. (Lines 196-199):

This replacement of $u_*$ with $u_{s*}$ in Eq. (13) makes the saltation equation consistent with the physics of aeolian transport in the presence of roughness, where excess wind friction speed at the soil surface (i.e., friction speed above the threshold required for particle mobilization) governs the saltation mass flux (Webb et al., 2020).

[193-195] – So whenever MODIS fails (missing data), Us* = 0 so there are no dust emissions in those pixels (or the whole domain if the retrieval fails broadly)? Why not default to the CTRL parameters if there is missing data?

Response: Thank you for raising this issue. We did not encounter any issues with missing data in our case study, but we can see the potential for missing data problems to occur. The reviewer is correct in that the equations would interpret missing data as $u_{s*} = 0$. Rather than defaulting to the control parameters, we suggest users fill the data gaps by interpolating nearby points (assuming the missing data gaps are minimal), using climatological $u_{ns*}$ values (something we've already suggested exploring in our discussion section to save on processing time), or using an input dataset from a previous time period.

We've added the following text to the end of Sect. 2.1.2 (Lines 207-210):

We note the potential for missing data problems if the MODIS retrievals have poor spatial coverage. If this scenario occurs, we recommend users consider filling the gaps through interpolation techniques (assuming the missing data gaps are minimal), using seasonal or monthly climatological $u_{ns*}$ values, or using a $u_{ns*}$ input dataset from a previous period.

[Section 2.3] – Is wet deposition of dust included? What about the convective transport of dust? These should also be added to Table 3.

Response: All relevant configuration settings are listed in Table 3. A copy of our entire configuration file (i.e., *namelist.input* file) is also available (see Michaels et al., 2022; Appendix B). Our study uses the Georgia Institute of Technology–Goddard Global Ozone Chemistry Aerosol Radiation and Transport (GOCART; Chin et al., 2000; Ginoux et al., 2001) "simple" modules as they are implemented in WRF-Chem v4.1. Once lofted, dust particles become a relatively passive "tracer" unless WRF-Chem aerosol feedback settings are activated (which we did not incorporate in our study to maintain consistency in forcing conditions across the simulations for all five test configurations). Vertical transport and time spent in suspension are primarily governed by the balance of simulated updrafts (including convective processes), atmospheric mixing, and dust deposition rates. Dust deposition in GOCART is mainly driven by gravitational settling and dry deposition, though GOCART will often remove dust from the atmosphere under rainy conditions even without the use of indirect feedback code adaptations (e.g., aerosol effects on modeled cloud microphysics and precipitation). While others have explored the use of more sophisticated wet deposition treatments with WRF-Chem/GOCART/AFWA (e.g., Tsarpolis et al., 2018), these modifications are not part of the standard WRF-Chem v4.1 baseline code distributed by NCAR.

We note that elevated atmospheric dust concentrations associated with the main dust event generally align with the location of the outflow boundary in all simulations. $PM_{10}$ concentrations immediately under the rain-producing convective cells are an order of magnitude or more lower (e.g., Fig. 12). While additional studies are necessary to fully resolve the issue, we suspect that the model improvements made through drag partition incorporation far outweigh simulation errors introduced by the general lack of wet deposition treatments, at least for this case.

[220] – 40 vertical levels is pretty coarse, especially for convective events. Since this is a cold pool case, how many levels are there in the boundary layer?

Response: Thank you for addressing this. Vertical resolution in the boundary layer, especially for convective events, is an essential component. While 40 vertical levels may initially sound coarse, our distribution of model levels ensured that each level was spaced no further than 1 km apart throughout the entirety of the atmosphere and placed 10 levels within 1 km above ground level. This was the default configuration of the WRF model v4.1 when we first performed our study and adhered to the suggested community guidelines. This compressed distribution of model levels closer to the surface allows our simulations to benefit from the improved vertical resolution where it is most essential for adequately resolving the temporally and spatially smaller intense vertical motions that contribute to the development of convective storms, while also reducing computational burden in the upper atmosphere where vertical motions are weaker and

larger in temporal and spatial scales. However, we acknowledge convective forecasts are sensitive to changes in vertical resolution. As such, Gallagher et al. (2022) investigated the effects of increasing the number of model levels to 65 for our case study event and found the improvement in forecast skill to be negligible, while the computational cost was notable.

[258] – How would one go about tuning Cs? Also, there are already tuning constants (C) in dust models. Why go through the extra steps and use Cs rather than the classic C tuning in the bulk flux equation? Either way the model is being tuned to some sort of observation.

Response: How to best tune $C_s$ or any of the other tuning constants in the AFWA module is an interesting question but is outside the scope of this study. While not discussed in this paper, the AFWA module does incorporate an optional "classic C" global tuning constant for the bulk vertical dust equation that users can set at run time (referred to as $c_\alpha$ in LeGrand et al., 2019). We note that tuning one versus the other will likely result in different patterns since the $C_s$ scaling parameter changes a thresholded cubic relationship, while $c_\alpha$ is a simple linear scaling approach. The extent to which those tuning approaches are meaningful is beyond the scope of this study.

[315] – The simulated reflectivity also produces less widespread precipitation than the observations. What if the high dust levels in the control case is from less rain leading to insufficient wet scavenging of dust and not from over-emission? Could you compare precipitation measurements to precipitation in the model?

Response: Thank you for bringing this to our attention. This is a valid criticism given our evaluation of the atmospheric forcing conditions did not directly involve comparison with precipitation measurements. Our atmospheric validation component, Gallagher et al. (2022), opted not to focus on precipitation verification due to the relatively low number and tight clustering of in-situ precipitation observations. We focused more on overall storm morphology than resulting precipitation, ensuring that convective phenomena and lofting winds were well represented. However, we wish to point out that the predominant phenomenon driving dust emission and transport, in this case, was the gust front ahead of the quasi-linear convective system. This surface boundary remained just ahead of the radar signatures for the event duration until it started to collapse and dissipate towards the end, approximately 4 July 2014, 0800 UTC. Also, with regards to the reflectivity spatial coverage, we recognize that direct comparisons of spatial extent come with the caveat that the observed radar is a national composite of multiple radar returns, stitched together from various heights above the surface, while our simulated reflectivity is consistently at 1 km above ground level and may occasionally represent different "slices" through the storm. Lastly, the overabundance of spurious dust emission in the CTRL simulation emanated from the area north of the Gulf of California. This particular area was well west of the main storm event and largely cloud/precipitation free in both the simulation and observation data.

[329-330] – Not sure what this comment about shrubs and grasses has to do with the point preceding it.

Response: We reworded this sentence to clarify (Lines 366-369):

We note that the simulated $u_*$ values are generally an order or magnitude stronger than their $u_{s*}$ partition. This outcome is likely due to the drag partitioning scheme interpreting the relatively "dark" terrain surfaces the domain landscape (presumably caused by prevalence of shrubs, grasses, or trees) as areas with substantial roughness element coverage.

[347] – The statement about soil moisture being important here contradicts the statement in [350-351] about it being relatively unimportant. It's probably just a wording issue. Note that Bukowski & van den Heever (2022) also found soil moisture to be relatively unimportant in haboobs.

Response: Thank you for the comment. We restructured this block of text to help clarify. Also, please note, this part of our assessment focuses on how air density and soil moisture moderate the wind friction speed threshold required for sediment mobilization ($u_{*ts}$), not the overall dust emission flux. This aspect of the model behavior is important because there are simulated $u_{ts*}$ values in the forest-covered mountain areas, which should not be generating dust, that are well below the simulated $u_*$ values. Once we remove the $z_0$ and vegetation masking elements of the original model configuration (i.e., once we shift to ALT3 and ALT4 configuration settings), the only component of the dust module preventing dust emissions from these areas is the drag partition treatment.

Lines 384-396:

===

Air density is the only spatiotemporally varying parameter in the calculation of $u_{*ts}(D_{s,p})$. Though we can discern a slight reduction in $u_{*ts}(D_{s,p} = 69\ \mu m)$ over time immediately under the convective line (Fig. 9m-p), the overall effect of air density on $u_{*ts}(D_{s,p})$ for this case is relatively negligible. Under air-dry soil conditions, $u_{*ts}(D_{s,p} = 69\ \mu m)$ ranges between 0.17 to 0.19 m $s^{-1}$ across most of the domain. These results also align with findings by Darmenova et al. (2009) in their assessment of the sensitivity of the Marticorena and Bergametti (1995) dust emission scheme to uncertainties in its required input parameters.

We also see relatively little change in the $f(\theta)$ field during the dust event, except for the area associated with a line of precipitation that occurred within the convective cell behind the main wall of dust (Fig. 9q-t). The $u_{*ts}(D_{s,p} = 69\ \mu m, \theta)$ maxima over the Mogollon Rim align with isolated areas of convective precipitation that occurred earlier in the simulation (Fig. 9u-x). Note, however, that the $u_{*ts}(D_{s,p} = 69\ \mu m, \theta)$ values along the Mogollon Rim adjacent to these maxima are around 0.2 to 0.3 m $s^{-1}$, comparable to $u_{*ts}(D_{s,p} = 69\ \mu m, \theta)$ values in the southwest Arizona region where the dust event occurred, and, for the most part, well below the simulated values of $u_*$. This particular aspect of the model behavior is important because the only component of the AFWA module preventing dust emissions from these drier forested areas in the ALT3 and ALT4 simulations is the drag partition treatment.

[470-471] – Terms.

Response: We rephrased these sentences; however, it was not entirely clear to us which terms the reviewer had concerns over.

Lines 517-522:

We note, however, that modeled 10 m wind speeds will be reduced over forested areas due to the aerodynamic drag from the trees on simulated wind speeds and that these particular aerodynamic roughness effects vary as a function of the $z_0$ settings in the parent WRF model. The drag partition treatment, at least with respect to how it's configured in our modeling set up, has no influence on simulated winds outside of the dust emission calculation. Even if the $u_{ns*}$ values were high for a model grid cell with forest cover, the simulated winds would likely be reduced by the internal WRF model physics, potentially mitigating (or masking) a shortfall in the drag partition correction.

[Table A1] – It would be great to have more in-depth descriptions of variables. E.g. rather than just calling something a "constant," describe what that constant represents. Also a column for units would help.

Response: Thank you for the suggestion. We added units where appropriate in terms of [L]=Length, [T]=Time, and [M]=Mass. Due to the size of the table, we believe the second column provides enough information to aid the reader in keeping track of the large number of parameters in this paper. The table is meant to serve as a "quick reference" guide for readers. Unfortunately, providing a glossary of in-depth descriptions would defeat this purpose and make the table prohibitively long. Readers are also able to review the AFWA module equation schematic in Fig. 6 to visualize how these parameters are used within the AFWA module.

We updated the appendix text to remind readers that Tables A1-A3 pair well with the schematic from Fig 6. to summarize the AFWA module parameters and equations.

Lines 626-629:

Tables A1-A3 provide the symbol, name, units, and value or description of variables referred to throughout this paper. Values of prescribed constants are listed. Variable arrays are described as "variable" for size bin-related settings, "spatially varying parameter" for static fields, and "spatiotemporally varying parameter" for temporally dynamic fields. See Fig. 6 for a schematic overview of how these parameters are used in the AFWA module calculations.

[Fig. 3] – Maybe add variables to plot in case readers forget the long description. E.g. Source Function (S).

Response: Thank you for this suggestion. We have updated the figures accordingly.

[Fig. 3] – The colorbar scale for sandblasting makes it look constant. Range of values may need to be adjusted to see heterogeneities.

Response: Please note, the sandblasting field is constant across most of the domain. We include this figure to bring awareness to the general lack of influence this parameter has on the bulk dust emission flux beyond serving as a scaling factor.

We've added the following commentary to the text and the Fig. 4 (formerly Fig. 3) caption to help make this clearer:

Fig. 4 caption: … Note that $\beta$ (panel f) is relatively homogenous due to the clay-rich soil content of the domain and is capped at $1.06 \times 10^{-6}$ $cm^{-1}$ where the soil composition exceeds 20% clay content. …

Lines 298-301: Note that $\beta$, in this case, is homogenous in the study area (e.g., Fig. 4f) due to the relatively clay-rich soil content of the region (e.g., Fig. 4b) and is capped at $1.06 \times 10^{-6}$ $cm^{-1}$ everywhere the soil composition exceeds 20% clay content (e.g., Fig. 4c). Accordingly, we can assume that the resultant simulated dust emission patterns produced by our test configurations are a function of $Q$, $S$, the $z_0$ mask, or some combination of these parameters.

[Fig. 4] – Panel a is tough to figure out with the overlap.

Response: Thank you for the suggestion. We've updated the legend on Fig. 5a (formerly Fig. 4a) to help clarify.

[Fig. 8] – Maybe it's the scaling of the colorbars again, but it is very difficult to see temporal changes in any of the variables.

Response: Thank you for the comment. We debated quite a bit about how to set the color bar gradients. The primary concept we want this figure to communicate is that the drag partition has more influence on the model results than any temporal change. If we were to change the color bars, that visual cue would be much less apparent.

[Fig. 11] – The colors in this colorbar are tough to discern since brown-orange represents low values and high values.

Response: Thank you for the comment. We understand the concern; however, we chose this color bar based on accepted guidance for colorblind-friendly palettes. The high and low gradient is easy to discern in the continuous spatial plots but can be a bit confusing in extreme edges of the domain for the station-based plot. Given that the macro patterns are discernable in areas relative to the discussion the figure was intended to support, we opted to keep the figure shading since the EPA station data are also publicly available for direct review.

- Added language to Eq. (10) description explaining how the albedo normalized by the BRDF weighting parameter, in theory, can be used to estimate a characteristic aerodynamic roughness for an area.
- Included information about how our modified code ingests preprocessed $u_{ns*}$ data through an auxiliary channel at model run time.
- Added suggestions for how to handle missing MODIS data issues.
- Clarified definitions and descriptions of several terms, including source strength ($S$), aerodynamic roughness length ($z_0$), the soil surface and roughness components of wind friction speed ($u_{s*}$ and $u_{r*}$, respectively), excess wind friction speed, and wind erosivity.
- Added commentary in the text and in the Fig. 4 caption on how the domain's relatively high soil clay content causes the sandblasting efficiency factor ($\beta$) used in the bulk vertical dust emission flux calculation to be relatively homogeneous in the study area.
- Added text to the beginning of the results section to provide readers with a "road map" of how the results are organized.
- Expanded the surface wind speed evaluation to include the quantitative wind speed bias analysis results from Gallagher et al. (2022).
- Clarified statements in the $u_{*\mathrm{ts}}$ component analysis results section to emphasize AFWA module functionality over relevance to dust modeling in general. Reviewing the $u_{*\mathrm{ts}}$ field and the role of the parameters feeding into it is important from a model mechanics perspective because there are simulated $u_{*\mathrm{ts}}$ values in the forest-covered mountain areas that are well below the simulated $u_*$ values. Under the ALT3 and ALT4 configuration settings, the only component of the dust module preventing dust emissions from these areas is the drag partition treatment.
- Added analysis on the contribution of aerosol species other than dust to the simulated $PM_{10}$ load. Dust was the primary source of simulated $PM_{10}$ over land.
- Clarified statement about the substantial influence of the albedo-based drag partitioning method on $u_{s*}$ occurring because the albedo scheme is picking up on the "dark" landscape areas covered by vegetation.
- Added commentary to the conclusion section on research and forecasting applications that could potentially benefit from incorporating satellite-retrieved drag partition information into the AFWA dust emission module.
- Updated the Appendix A tables to include units and new parameters.
- Updated the model code availability section. We updated and released a new version of our code with a bug fix. The new version (v1.1.1) is nearly identical to the v1.1.0 release except for a typo correction in the AFWA dust emission module (line 162 in the code).

This error was causing the drag-partitioned modified code to set ustar to 0 when running with the default AFWA dust emission module configuration settings. This typo, however, did not affect our study because it was incorporated by mistake after we had already completed the CTRL simulation used in our study. We also added full copies of our WRF-Chem configuration files to the repository.

- Added a few references suggested by the reviewers.
- Corrected a few minor misspellings, duplicate words, and punctuation errors.